# Optimizing Decoding Paths in Masked Diffusion Models by Quantifying Uncertainty

## Abstract

Masked Diffusion Models (MDMs) offer flexible, non-autoregressive generation, but this freedom introduces a challenge: final output quality is highly sensitive to the decoding order. We are the first to formalize this issue, attributing the variability in output quality to the cumulative predictive uncertainty along a generative path. To quantify this uncertainty, we introduce Denoising Entropy, a computable metric that serves as an internal signal for evaluating generative process. Leveraging this metric, we propose two algorithms designed to optimize the decoding path: a post-hoc selection method and a real-time guidance strategy. Experiments demonstrate that our entropy-guided methods significantly improve generation quality, substantially boosting accuracy on challenging reasoning, planning, and code benchmarks. Our work establishes Denoising Entropy as a principled tool for understanding and controlling generation, effectively turning the uncertainty in MDMs from a liability into a key advantage for discovering high-quality solutions.

## 1 Introduction

In language modeling, Masked Diffusion Models (MDMs) (Austin et al., 2021a; Lou et al., 2023; Sahoo et al., 2024) are rapidly emerging as powerful counterparts to the dominant Auto-Regressive Models (ARMs). While ARMs rely on the next-token prediction paradigm and are constrained to a fixed left-to-right generation path (Bengio et al., 2003; Sutskever et al., 2014), MDMs learn to denoise sequences via random-order masking, enabling them to construct sequences in any order in principle. Each unique sequence construction process can be viewed as a *decoding path*, and different paths typically entail varying generation difficulty that shapes the final output. The generation flexibility of MDMs thus has a theoretical potential: within the vast space of possible decoding paths, there must exist an optimal path that yields a results no worse than the single, rigid sequential path of ARMs.

However, this theoretical potential often remains untapped in practice, as MDMs always underperform compared to ARMs (Feng et al., 2025). Beyond the model capability, decoding strategy acts as a critical bottleneck determining the path taken (Kim et al., 2025). The default random-order strategy, for instance, treats all paths as equally probable (Austin et al., 2021a), reducing the search for an optimal path to mere chance. More sophisticated approaches employ greedy token-level heuristics, such as unmasking the token with the highest confidence (Chang et al., 2022), but are still myopic: a sequence of locally optimal steps provides no guarantee of a globally optimal path. The fundamental limitation underlying these strategies is their lack of a global perspective, as they operate without a mechanism to assess the quality of the overall generative path.

We draw inspiration from uncertainty quantification in ARMs, where metrics like entropy reliably link predictive uncertainty to generative quality (Xu et al., 2020; Kuhn et al., 2023), and uncertainty-aware decoding has shown effect for enhancing global coherence (Arora et al., 2023; Zhu et al., 2024). Adapting this insight to MDMs, we formalize **Path Uncertainty**: MDM's cumulative predictive uncertainty along a complete decoding path, capturing the global quality of the entire generative process. We attribute MDM quality variability to path uncertainty. High cumulative uncertainty harms output consistency, while low uncertainty indicates reliable paths (Press et al., 2024).

We introduce **Denoising Entropy**, a novel and internally computable metric designed to quantify the inherent uncertainty of generative paths in MDMs. As illustrated in Figure 1, Denoising Entropy comprises two complementary variants: State Entropy ($h_{\mathrm{DE}}$), which captures uncertainty at a single

Figure 1: **Quantifying Path Uncertainty in MDMs with Denoising Entropy.** **Left**: State Entropy ($h_{\text{DE}}$) measures per-state uncertainty, computed as the mean Shannon Entropy over the predictive distributions for all masked positions. $h_{\text{DE}}$ is then aggregated over the entire path to form the Path Entropy ($H_{\text{DE}}$). **Right**: Decoding process shows how different paths yield outputs of varying quality. Our key insight is that the lower $H_{\text{DE}}$ indicates path yeilding better output, providing a potent internal signal for generation quality.

decoding state, and Path Entropy ($H_{\text{DE}}$), which integrates this uncertainty across an entire trajectory to measure cumulative generative uncertainty. This formulation provides a principled framework for evaluating the reliability of different generative paths.

Minimizing Denoising Entropy offers a powerful new objective for steering MDM decoding towards more reliable outcomes. We operationalize this principle through two search algorithms: a post-hoc selection method that reranks sampled paths to find the one with the minimum Path Entropy ($H_{\text{DE}}$), and a real-time guidance method that uses State Entropy ($h_{\text{DE}}$) to actively steer generation towards lower-uncertainty regions of the state space. By shifting the focus from myopic, token-level decisions to holistic, path-level optimization, our approach significantly boosts the performance of MDMs on complex open-ended generation and reasoning tasks. **We summarize our key findings below:**

- **Formalizing Path Uncertainty** (Section 3.1): We identify and formalize Path Uncertainty, the cumulative predictive uncertainty of a MDM along a single decoding path, which drives the variability in output quality.
- **Denoising Entropy as Uncertainty Proxy** (Section 3.2): We introduce two novel and computable metrics, $h_{\text{DE}}(t)$ and $H_{\text{DE}}$, as a theoretically-grounded toolkit to evaluate entire generative states and paths, providing signals for path-aware guidance.
- **Path Search Algorithms** (Section 3.3): We propose two path search algorithms, Entropy-based Best-of-N and Entropy-guided Sequential Monte Carlo, that leverage Denoising Entropy as an internal signal to optimize the decoding process and improve the quality of generated sequences.

## 2 PRELIMINARIES

Let $\mathcal{V}$ be a vocabulary of size $K$, and consider a sequence of $L$ tokens $\mathbf{x}_0 = (x_0^1, \ldots, x_0^L)$. For any position $\ell \in \{1, \ldots, L\}$, the token $x_0^\ell$ is represented by a one-hot vector $\mathbf{x}_0^\ell \in \{0, 1\}^K$, whose component corresponding to the index of $x_0^\ell$ is 1.

### 2.1 FORWARD PROCESS AND TRAINING OBJECTIVE

The training of MDMs relies on a continuous-time forward corruption process. For any time $t \in [0, 1]$, a latent variable $\mathbf{z}_t$ is generated from $\mathbf{x}_0$. This is achieved probabilistically: each token $x_0^\ell$ is independently replaced by a special [MASK] token with probability $1 - \alpha_t$, and remains unchanged with probability $\alpha_t$. There $\alpha_t \in [0, 1]$ is a strictly decreasing noise schedule with $\alpha_0 \approx 1$ and $\alpha_1 \approx 0$ (Sahoo et al., 2024). Let $\mathcal{M}_t$ be the set of indices that are masked at time $t$.

An MDM with denoising network $p_{\boldsymbol{\theta}}$, parameterized by $\boldsymbol{\theta}$, is trained to reverse this process. Given the corrupted variable $\mathbf{z}_t$ and the time $t$, the MDM predicts the probability distribution $\hat{\mathbf{p}}_0$ for each token in the original sequence as an estimate of $\mathbf{x}_0$:

$$\hat{\mathbf{p}}_t^\ell = p_{\boldsymbol{\theta}}(X_t^\ell | \mathbf{z}_t, t) \in \Delta^K, \quad \forall \ell \in \{1, \ldots, L\}, \tag{1}$$

where $X_t^\ell$ is the random variable for the token at position $\ell$ and time $t$, $\Delta^K$ is the $K$-simplex.

Parameters $\boldsymbol{\theta}$ are optimized by minimizing the Negative Evidence Lower Bound (NELBO)(Lou et al., 2023; Shi et al., 2024; Sahoo et al., 2024). This objective simplifies to an integral of the weighted negative log-likelihood over the masked tokens:

$$\mathcal{L}(\boldsymbol{\theta}) = \mathbb{E}_{\mathbf{x}_0 \sim q(\mathbf{x}_0)} \left[ \int_0^1 w(t) \mathbb{E}_{\mathbf{z}_t | \mathbf{x}_0} \Big[ \sum_{\ell \in \mathcal{M}_t} -\log p_{\boldsymbol{\theta}}(x_0^\ell | \mathbf{z}_t, t) \Big] dt \right], \tag{2}$$

where $q(\mathbf{x}_0)$ is the posterior data distribution, $1 - \alpha_t$ is the unmasking probability at time $t$ and $w(t) = \left| \frac{d\alpha_t}{dt} \right| \frac{1}{1 - \alpha_t}$ is the weighting function.

## 2.2 REVERSE PROCESS FOR GENERATION

The inference process generates a sequence $\mathbf{x}_0$ by iteratively denoising from a fully masked sequence. The procedure starts with $\mathbf{z}_1$, where all tokens are [MASK]. It then proceeds backward in time along a discrete schedule $1 = t_N > t_{N-1} > \cdots > t_0 = 0$.

At each step $i$ (from $N$ down to 1), the MDM $p_{\boldsymbol{\theta}}$ takes the current state $\mathbf{z}_{t_i}$ and time $t_i$ as input to predict the original token probability distribution $\hat{\mathbf{p}}_0^\ell$ for each position $\ell$. Then, the next state $\mathbf{z}_{t_{i-1}}$ is sampled from the reverse transition kernel $p(\mathbf{z}_{t_{i-1}} | \mathbf{z}_{t_i})$. This distribution allows a masked token to either be unmasked to a content token (sampled according to $\hat{\mathbf{p}}_0^\ell$) or to remain masked, governed by the noise levels $\alpha_{t_i}$ and $\alpha_{t_{i-1}}$. This iterative refinement continues until $t_0 = 0$, yielding the final generated sequence.

# 3 MODELING PATH UNCERTAINTY WITH DENOISING ENTROPY

## 3.1 PATH UNCERTAINTY

A decoding path $\tau = (\mathbf{z}_{t_N}, \ldots, \mathbf{z}_{t_0})$ is a sequence of latent states traversed during the generative process, guided by a decoding strategy $\pi$. The strategy $\pi$ determines the transition from $\mathbf{z}_{t_i}$ to $\mathbf{z}_{t_{i-1}}$ at each step, based on the model transition kernel $p(\cdot | \cdot)$.

We define **Path Uncertainty** as the model cumulative predictive uncertainty along a *single* decoding path $\tau$, which reflects how *confident* the model is throughout the entire generation of a sequence.

## 3.2 DENOISING ENTROPY

To quantify path uncertainty, we introduce **Denoising Entropy**. We first define *Oracle State Uncertainty* as the output uncertainty of the joint prediction distribution at any given state.

> **Definition 1 (Oracle State Uncertainty) .** *Let* $\mathbf{X}_{\mathcal{M}_t} = \{X_0^\ell\}_{\ell \in \mathcal{M}_t}$ *be the random vector of the true tokens at all* [MASK] *positions. Oracle State Uncertainty is the Shannon entropy of the model's joint predictive distribution over this vector, conditioned on the latent state* $\mathbf{z}_t$:
>
> $$H_{oracle}(\mathbf{z}_t) \triangleq H(\mathbf{X}_{\mathcal{M}_t} | \mathbf{z}_t). \tag{3}$$

Then, the *State Entropy* is defined to measure the average instantaneous prediction uncertainty.

> **Definition 2 (State Entropy) .** *For a given latent sequence* $\mathbf{z}_t$ *at time* $t \in [0, 1]$*, State Entropy* $(h_{DE})$ *is defined as the average Shannon entropy of the model predictive distributions over the set of masked positions* $\mathcal{M}_t$:
>
> $$h_{DE}(\mathbf{z}_t) \triangleq \frac{1}{|\mathcal{M}_t|} \sum_{\ell \in \mathcal{M}_t} H\left(p_{\boldsymbol{\theta}}(X_0^\ell | \mathbf{z}_t, t)\right), \tag{4}$$
>
> *where* $H(\cdot)$ *denotes the Shannon entropy and the definition is valid for* $|\mathcal{M}_t| > 0$.

The integral of *State Entropy* across the entire path then yields *Path Entropy*, which quantifies the cumulative uncertainty of the generative process.

**Definition 3 (Path Entropy).** *Path Entropy ($H_{DE}$) provides measure for the total uncertainty of a denosing process with a decoding path $\tau$. It is the integral of the $h_{DE}$ over the generation time. In a discrete-time setting with $N$ steps, it can be approximated by averaging the $h_{DE}$ across all steps:*

$$H_{DE}(\tau) \triangleq \int_0^1 h_{DE}(\mathbf{z}_t)dt \approx \frac{1}{N}\sum_{i=1}^{N} h_{DE}(\mathbf{z}_{t_i}). \qquad (5)$$

### 3.3 ENTROPY-GUIDED DECODING

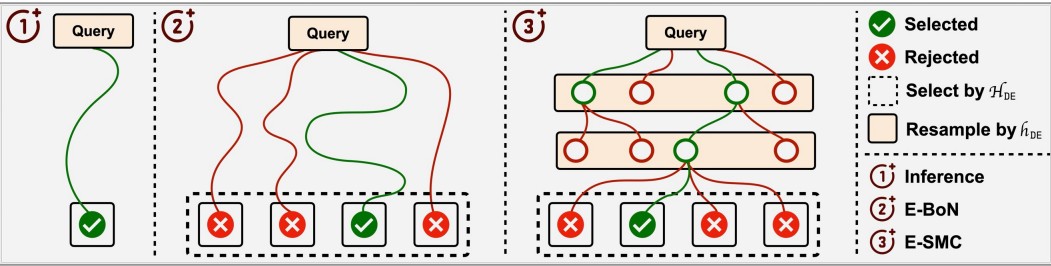

Figure 2: **An overview of our entropy-guided decoding algorithms.** While *standard inference* in subfigure (1) generates a single decoding path, our methods explore multiple candidate paths guided by Denoising Entropy. **E-BoN** in subfigure (2) performs **post-hoc selection**, choosing the best path from multiple independent candidates based on path entropy $H_{DE}$. **E-SMC** in subfigure (3) provides **real-time guidance**, iteratively pruning high-entropy paths and replicating low-entropy ones based on state entropy $h_{DE}$ of partial solutions.

A decoding strategy $\pi$ induces a distribution $P(\tau|\pi)$ over paths and their corresponding path entropies $H_{DE}(\tau)$. This implies two properties: (i) a single stochastic strategy yields a variance in path uncertainty, $\mathbb{V}_{\tau \sim P(\cdot|\pi)}[H_{DE}(\tau)] > 0$; and (ii) distinct strategies, $\pi_a$ and $\pi_b$, produce different expected uncertainties, $\mathbb{E}_{\pi_a}[H_{DE}(\tau)] \neq \mathbb{E}_{\pi_b}[H_{DE}(\tau)]$.

Hypothesizing that low path uncertainty lead to higher-quality outputs (Xu et al., 2020), generative decoding can be framed as an optimization problem. For a generative task, the objective is to find an optimal strategy $\pi^\star$ that minimizes the expected path entropy:

$$\pi^\star = \arg\min_\pi \mathbb{E}_{\tau \sim P(\cdot|\pi)}[H_{DE}(\tau)]. \qquad (6)$$

While finding the globally optimal strategy $\pi^\star$ is intractable due to the vast search space, we introduce two algorithms that leverage denoising entropy as an active guidance signal: (i) *a post-hoc selection method, **Entropy-based Best-of-N** (**E-BoN**)*, and (ii) *a real-time path optimization strategy, **Entropy-guided Sequential Monte Carlo** (**E-SMC**)*, both designed to effectively approximate this objective. Figure 2 provides a conceptual overview E-BoN, E-SMC, and standard single-path inference.

#### 3.3.1 ENTROPY-BASED BEST-OF-N

$H_{DE}$ can be used for post-hoc selection from a set of candidates. E-BoN formalizes this principle. Given a population of $M$ candidate decoding paths, $\{\tau^{(1)}, \ldots, \tau^{(M)}\}$, E-BoN identifies the single best path with minimum $H_{DE}$:

$$\tau^\star = \arg\min_{m \in \{1, \ldots, M\}} H_{DE}(\tau^{(m)}). \qquad (7)$$

E-BoN is straightforward, requiring the full generation of $M$ samples before selection. However, it expends the full computational budget uniformly across all paths, without any mechanism to redirect resources during generation (Chatterjee & Diaconis, 2018).

#### 3.3.2 ENTROPY-GUIDED SEQUENTIAL MONTE CARLO

To actively optimize the decoding path in real-time, we introduce **E-SMC**, a variant of Sequential Monte Carlo (Doucet et al., 2001) adapted for MDM reverse process (Singhal et al., 2025). E-SMC employs a guidance mechanism that operates on $M$ parallel particles. By evaluating each partial

---

**Algorithm 1** Entropy-guided Sequential Monte Carlo

---

1: **Input:** MDM $\theta$, number of particles $M$, temperature $\lambda$, resampling interval $\Delta i_r$.
2: **Initialize:** Sample initial particle set $\mathcal{Z}_{t_N} = \{\mathbf{z}_{t_N}^{(m)}\}_{m=1}^M$ where $\mathbf{z}_{t_N}^{(m)} \sim p(\mathbf{z}_{t_N})$.
3: **for** $i = N, N-1, \ldots, 1$ **do**
4:    **Propagate:** For each particle $m \in \{1, \ldots, M\}$, sample $\mathbf{z}_{t_{i-1}}^{(m)} \sim p_\theta(\cdot|\mathbf{z}_{t_i}^{(m)})$.
5:    **if** $(N - i + 1) \pmod{\Delta i_r} = 0$ **and** $i > 1$ **then**
6:       **Evaluate:** For each particle $m$, compute potential $G^{(m)} \leftarrow \Phi(\mathbf{z}_{t_{i-1}}^{(m)}; \lambda)$.
7:       Normalize to get probabilities: $w^{(m)} \leftarrow G^{(m)} / \sum_{j=1}^M G^{(j)}$ for all $m$.
8:       **Resample:** Draw ancestor indices $\{a^{(m)}\}_{m=1}^M$ where $a^{(m)} \sim \text{Categorical}(\{w^{(j)}\}_{j=1}^M)$.
9:       Update particle set: $\mathcal{Z}_{t_{i-1}} \leftarrow \{\mathbf{z}_{t_{i-1}}^{(a^{(m)})}\}_{m=1}^M$.
10:   **end if**
11: **end for**
12: **Return:** Final population $\mathcal{Z}_{t_0}$.

---

path via State Entropy $h_{\text{DE}}$, E-SMC can dynamically reallocate computational budget and prune high-entropy paths while replicating low-entropy ones. This iterative search process is structured around three steps: *Propagation, Evaluation, and Resampling:*

- **Propagation.** At each reverse step $i \in \{N, \ldots, 1\}$, we advance each particle $\mathbf{z}_{t_i}^{(m)}$ in the set $\mathcal{Z}_{t_i}$ to a new state $\mathbf{z}_{t_{i-1}}^{(m)}$ by sampling from the proposal distribution given by MDM reverse kernel, $\mathbf{z}_{t_{i-1}}^{(m)} \sim p(\cdot|\mathbf{z}_{t_i}^{(m)})$. This process generates the subsequent particle population $\mathcal{Z}_{t_{i-1}}$.

- **Evaluation.** To actively guide the search, evaluation are performed periodically at fixed intervals of $\Delta i_r$ steps. During evaluation, each particle $\mathbf{z}_{t_{i-1}}^{(m)}$ in current set $\mathcal{Z}_{t_{i-1}}$ is assessed using a potential function $\Phi(\mathbf{z}; \lambda)$. $\Phi$ is designed to assign a higher score to particles with lower $h_{\text{DE}}(\mathbf{z})$. Temperature $\lambda$ modulates the sharpness of the resulting score distribution. $\Phi$ is detailed in Appendix A.

- **Resampling.** Potential scores computed during evaluation are normalized to form a categorical distribution over the current particle set $\mathcal{Z}_{t_{i-1}}$. This distribution assigns a selection probability $w^{(m)}$ to each particle. A new population is formed by drawing $M$ particles with replacement from $\mathcal{Z}_{t_{i-1}}$ according to $w^{(m)}$. Resampling mechanism prunes high-entropy paths while replicating low-entropy ones, steering search towards more promising path space (Moral, 2004). Full process is detailed in Algorithm 1.

### 3.4 THEORETICAL JUSTIFICATION FOR DENOISING ENTROPY

The effectiveness of E-BoN and E-SMC, is predicated on Denoising Entropy being a indicator for generation quality. This section validates the use of this metric as a decoding objective by establishing its theoretical basis through two key properties. First, we prove that State Entropy $h_{\text{DE}}$ is a computable upper bound on the ideal, yet intractable, joint entropy of the masked tokens (Proposition 1). Second, we relate $h_{\text{DE}}$ directly to the model's training objective, showing it serves as a close proxy for the instantaneous loss (Proposition 2). Together, these results justify minimizing Denoising Entropy as a principled strategy for guiding the generative process toward more consistent and higher-quality outputs. We also provide a further analysis of Denoising Entropy properties in Appendix D.

**$h_{\text{DE}}$ as a bound on ideal uncertainty.** $h_{\text{DE}}$ is a well-posed measure of uncertainty by relating it to an ideal but intractable metric, *Oracle State Uncertainty* $H_{\text{oracle}}$, defined as the joint entropy over all [MASK] tokens. $h_{\text{DE}}$ serves as a computable upper bound to this oracle.

> **Proposition 1 ($h_{\text{DE}}$ as an upper bound).** *Oracle State Uncertainty $H_{oracle}(\mathbf{z}_t)$ is upper-bounded by the sum of marginal entropies, which is directly proportional to State Entropy $h_{DE}(\mathbf{z}_t)$:*
>
> $$H_{oracle}(\mathbf{z}_t) \leq \sum_{\ell \in \mathcal{M}_t} H\left(p_{\boldsymbol{\theta}}(X_0^\ell|\mathbf{z}_t, t)\right) = |\mathcal{M}_t| \cdot h_{DE}(\mathbf{z}_t). \qquad (8)$$

$h_{\mathrm{DE}}$ sums the uncertainty of each masked token independently, thereby ignoring contextual constraints that tokens impose on each other and forming a valid upper bound. Proof relying on the subadditivity of Shannon entropy (Shannon, 1948) is provided in Appendix B.1.

**$h_{\mathbf{DE}}$ as a proxy for MDM loss.** We define $\epsilon$-accurate condition with the step-wise error.

> **Definition 4 ($\epsilon$-Accurate Denoising Model).** *We say an MDM is $\epsilon$-accurate if, for any time step $t_i$, the Kullback-Leibler(KL) divergence between the true posterior transition kernel $q$ (conditioned on ground truth) and the model's reverse kernel $p_\theta$ is bounded:*
>
> $$\mathbb{E}_{\mathbf{z}_{t_i} \sim q}\left[ D_{\mathrm{KL}}\Big(q(\mathbf{z}_{t_{i-1}}|\mathbf{z}_{t_i}) \,\|\, p_\theta(\mathbf{z}_{t_{i-1}}|\mathbf{z}_{t_i})\Big)\right] \le \epsilon\,, \tag{9}$$
>
> *where $\epsilon$ is a small constant reflecting the model's training loss.*

Under the assumption of $\epsilon$-accuracy, $h_{\mathrm{DE}}$ serves as a direct proxy to approximate the instantaneous per-token training loss with error of $\mathcal{O}(\epsilon)$. We state the approximation using state entropy as follows:

> **Proposition 2 (Approximation of Normalized Loss by $h_{\mathbf{DE}}$).** *Assume that MDM is $\epsilon$-accurate. $h_{DE}(\mathbf{z}_t)$ provides an approximation to the instantaneous, per-token normalized training loss:*
>
> $$\frac{1}{|\mathcal{M}_t|} \sum_{\ell \in \mathcal{M}_t} \left( -\log p_{\boldsymbol{\theta}}(x_0^\ell|\mathbf{z}_t, t) \right) - h_{DE}(\mathbf{z}_t) \le \epsilon + \sqrt{\frac{\epsilon}{2}}logK\,, \tag{10}$$
>
> *where $K$ is the vocabulary set size.*

Proof in Appendix B.2. While $h_{\mathrm{DE}}$ measures instantaneous difficulty, $H_{\mathrm{DE}}$ extends by integrating it throughout the path, measuring total generation challenge and reflecting distribution uncertainty.

**Entropy gap as output quality bound.** We denote the true and model's path distribution as $\mathrm{Pr}$ and $\widehat{\mathrm{Pr}}$ respectively for decoding path $\tau$. Let $\mu_{\mathrm{Pr}} = \mathbb{E}_{\tau \sim \mathrm{Pr}}[H_{\mathrm{DE}}(\tau)]$, $\mu_{\widehat{\mathrm{Pr}}} = \mathbb{E}_{\tau \sim \widehat{\mathrm{Pr}}}[H_{\mathrm{DE}}(\tau)]$ be the expected Path Entropy evaluated on reference paths and model's generations, respectively.

> **Proposition 3 (Path Entropy Gap as a Quality Bound).** *Assume that $|H_{DE}(\tau)| \le B$ where $B$ is a positive constant. The KL divergence between the reference path distribution $\mathrm{Pr}$ and the generated one $\widehat{\mathrm{Pr}}$ is lower-bounded by the squared error in their expected Path Entropies:*
>
> $$D_{\mathrm{KL}}(\mathrm{Pr}\,\|\widehat{\mathrm{Pr}}) \ge \frac{1}{2B^2}\left(\mu_{\widehat{\mathrm{Pr}}} - \mu_{\mathrm{Pr}}\right)^2\,. \tag{11}$$

Making $\mu_{\widehat{\mathrm{Pr}}}$ closer to $\mu_{\mathrm{Pr}}$ is necessary for lower KL divergence, which represents higher output quality. Our algorithm E-BoN and E-SMC both reduce the path entropy gap $|\mu_{\widehat{\mathrm{Pr}}} - \mu_{\mathrm{Pr}}|$, tightening the lower bound for the KL divergence. Details in Appendix C.2 and C.3.

## 4 EXPERIMENTS

This section empirically studies Path Uncertainty and Denoising Entropy in MDMs through three stages: **(i)** validating Denoising Entropy as a quality-aligned internal metric on MDMs; **(ii)** improving generation by using E-BoN and E-SMC to optimize decoding paths; **(iii)** scaling our method to larger models and complex reasoning and planning tasks.

### 4.1 VALIDATING DENOISING ENTROPY AS A QUALITY METRIC

We study whether the internal metric $H_{\mathrm{DE}}$, computed online during generation as a proxy for path uncertainty, aligns with external text quality evaluations.

**Experimental setting.** We use a MDLM trained on OpenWebText (Gokaslan et al., 2019) with approximately 130 million non-embedding parameters for evaluation (Sahoo et al., 2024). The MDLM generates sequences of length 1,024 under random-order unmasking. We vary the number of denoising steps $S = 2^i$ for $i \in [5, 10]$ and use GPT2-Large (Radford et al., 2019), a substantially

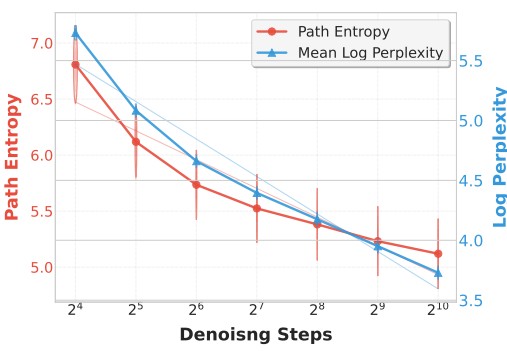 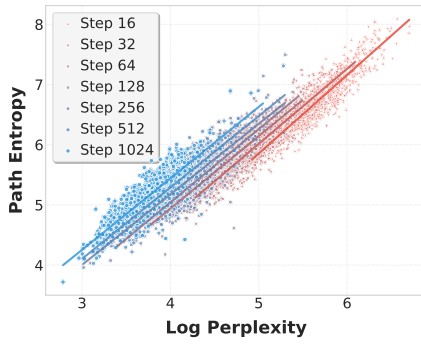

(a) Trend of $H_{\mathrm{DE}}$ vs. $\ln(\mathrm{PPL})$ with Denoising Steps.   (b) Per-sample correlation between $H_{\mathrm{DE}}$ and $\ln(\mathrm{PPL})$.

Figure 3: **Empirical validation of Path Entropy ($H_{\mathbf{DE}}$) as an internal proxy for generation quality.** The aggregate trend in **(a)** shows that a more refined generation process reduces both the model's internal uncertainty ($H_{\mathrm{DE}}$, red) and the output perplexity ($\ln(\mathrm{PPL})$, blue). **(b)** provides a per-sample analysis, revealing a strong positive linear correlation between $H_{\mathrm{DE}}$ and $\ln(\mathrm{PPL})$. Details in Appendix E.3.

larger model than the MDLM we used, as ARM evaluator to compute Perplexity (PPL). Perplexity is a standard evaluation metric for generative uncertainty and text quality quantification. For each generated sample, we record $H_{\mathrm{DE}}$ and $\ln(\mathrm{PPL})$.

**Results.** *Across thousands of unconditional generations, we observe a clear, near-linear relationship between $H_{DE}$ and $\ln(PPL)$ as shown in Figure 3.* Lower $H_{\mathrm{DE}}$ aligns with lower perplexity, indicating outputs with higher quality. Both $H_{\mathrm{DE}}$ and $\ln(\mathrm{PPL})$ decrease as the number of denoising steps $S$ increases, reflecting a reduction in path uncertainty and a improvement in sample quality. We use this case to show that *$H_{DE}$ is a reliable internal proxy for MDM-generated text quality*: from its own denoising dynamics, MDM can anticipate whether it is on a high- or low-quality path.

### 4.2 IMPROVING GENERATION BY USING DENOISING ENTROPY TO GUIDE PATH SEARCH

Building upon finding that $H_{\mathrm{DE}}$ serves as a internal proxy for text quality, we now investigate whether Denoising Entropy can be actively employed to mitigate Path Uncertainty and improve generation.

**Experimental setting.** We evaluate and compare three decoding types: vanilla single path uniform sampling, with E-BoN and E-SMC. We utilize the same MDLM as in Section 4.1, while adopting `GPT2-Large` and `Llama-3-8B` as the ARM evaluators. We analyze performance measured with perplexity and diversity (Zheng et al., 2024) across different numbers of denoising steps ($S$), particles ($K$), and resampling intervals ($\Delta i_r$). In detail:

- Diversity is used to assess whether repetitive degeneration occurs in generated content. For a sequence of length $L$ that contains $K$ distinct tokens, with each token $k$ occurring $L_k$ times. Diversity $D$ is computed as $D = -\sum_{k=1}^{K} \frac{L_k}{L} \log \frac{L_k}{L}$.
- For E-BoN, we generate $K$ independent particles and return the one with the minimum $H_{\mathrm{DE}}$.
- For E-SMC, we steer $K$ particles using weighted resampling with $h_{\mathrm{DE}}$ at intervals of $\Delta i_r$ steps, and finally return the one with the minimum $H_{\mathrm{DE}}$.
- We also implemented an entropy minimization method based on Greedy Search as a comparison to demonstrate the negative impact of over-optimization. Results are presented in Table 6.

**Results.** As shown in Table 1, both entropy-guided strategies substantially outperform the vanilla sampler, *confirming the practical utility of Denoising Entropy as a control signal.* E-SMC *consistently achieves the best performance, underscoring the benefit of its active, online guidance mechanism.* E-BoN *and* E-SMC *reduce perplexity while maintaining diversity comparable to the vanilla sampler. Under specific configurations,* E-SMC *can improve generation quality without compromising diversity. In contrast, other methods achieve significantly lower perplexity only by sacrificing diversity.*

Table 1: **Comparison of decoding strategies via ablation studies on hyperparameters.** The table analyzes the impact of the number of denoising steps ($S$), number of particles ($K$), and resampling interval ($\Delta i_r$). Perplexity is evaluated on two models: `GPT2-Large` and `Llama-3-8B`. E-SMC consistently shows the best performance, which can be achieved without compromising diversity in certain configurations.

| Configuration | | | Perplexity ↓ | | | Diversity ↑ | | |
|---|---|---|---|---|---|---|---|---|
| $S$ | $K$ | $\Delta i_r$ | Vanilla | E-BoN | E-SMC | Vanilla | E-BoN | E-SMC |
| 128 | 4 | 32 | 85.3/84.5 | 66.4/66.1 | **63.4/61.9** | 5.53 | 5.47 | 5.48 |
| 256 | 4 | 32 | 68.5/66.9 | 52.3/55.2 | **47.7/51.5** | 5.45 | 5.39 | 5.40 |
| 128 | 4 | 32 | 85.3/84.5 | 66.4/66.1 | **63.4/61.9** | 5.53 | 5.47 | 5.48 |
| 128 | 8 | 32 | 85.3/84.5 | 60.3/59.8 | **56.0/59.0** | 5.53 | 5.42 | 5.43 |
| 256 | 2 | 32 | 68.5/66.9 | 60.0/59.1 | **55.5/57.0** | 5.45 | 5.43 | 5.43 |
| 256 | 4 | 32 | 68.5/66.9 | 52.3/55.2 | **47.7/51.5** | 5.45 | 5.39 | 5.40 |
| 256 | 8 | 32 | 68.5/66.9 | 47.5/51.1 | **40.4/45.1** | 5.45 | 5.37 | 5.31 |
| 256 | 4 | 8 | 68.5/66.9 | 52.3/55.2 | **43.9/44.4** | 5.45 | 5.39 | 5.31 |
| 256 | 4 | 16 | 68.5/66.9 | 52.3/55.2 | **45.1/48.1** | 5.45 | 5.39 | 5.35 |
| 256 | 4 | 32 | 68.5/66.9 | 52.3/55.2 | **47.7/51.5** | 5.45 | 5.39 | 5.40 |
| 256 | 4 | 64 | 68.5/66.9 | 52.3/55.2 | **50.2/51.8** | 5.45 | 5.39 | 5.42 |
| 256 | 4 | 128 | 68.5/66.9 | 52.3/55.2 | **53.4/54.1** | 5.45 | 5.39 | 5.46 |

The results affirm the scalability of our methods, as performance improves directly with the computational budget. Increasing the number of particles ($K$) significantly enhances outcomes, particularly for E-SMC, which better exploits the expanded search space (e.g., perplexity drops from 47.7 to 36.1 as $K$ increases from 4 to 12). Furthermore, for E-SMC, more frequent resampling (a smaller $\Delta i_r$) provides a complementary and computationally efficient lever for improvement by actively pruning high-uncertainty paths.

## 4.3 SCALING ENTROPY GUIDANCE TO LARGE-SCALE REASONING TASKS

Experiments in Section 4.1 & 4.2 validated Denoising Entropy as a proxy for generation quality and demonstrated the efficacy of our proposed algorithms, E-BoN and E-SMC, on a small-scale MDM using perplexity as the evaluation metric. To assess the generalizability and practical impact of our approach, we now scale to advanced MDMs and evaluate on a suite of challenging reasoning benchmarks. This shift enables evaluation beyond abstract quality to task-specific accuracy, testing whether entropy guidance improves problem-solving in large-scale MDMs.

**Models and datasets.** We conduct experiments on three large MDMs: `LLaDA-8B-Instruct` (Nie et al., 2025), `LLaDA-1.5-8B` (Zhu et al., 2025), and `Open-dCoder-0.5B` (Peng et al., 2025b). `LLaDA` models are evaluated across five diverse benchmarks spanning three reasoning domains: mathematical reasoning (GSM8K (Cobbe et al., 2021), MATH500 (Lightman et al., 2024)), scientific reasoning (GPQA (Rein et al., 2024)), and planning (Sudoku (Nolte et al., 2024) and Countdown (Ye et al., 2025a)). `Open-dCoder-0.5B` is an MDM for code generation, we evaluate its performance on HumanEval/HumanEval+ (Chen et al., 2021) and MBPP/MBPP+ (Austin et al., 2021b).

**Decoding strategies.** We evaluate our path search algorithms with a spectrum of established MDM decoding strategies. These include the standard uniform sampler (Austin et al., 2021a), uncertainty-based samplers (e.g., confidence (Chang et al., 2022), entropy (Ben-Hamu et al., 2025), and margin (Kim et al., 2025)), efficient samplers (Semi-AR (Nie et al., 2025), EB-Sampler (Ben-Hamu et al., 2025), and Fast-dLLM (Wu et al., 2025)), and two recently proposed advanced strategies: P2 (Peng et al., 2025a) and PC-Sampler (Huang et al., 2025a). For block-wise strategy including Semi-AR and Fast-dLLM, we set the number of decoding blocks to 8 for all datasets. A detailed description of each baseline strategy is provided in Appendix E.1.

**Results: path search algorithms enhance decoding strategies.** We demonstrate the efficacy of E-BoN and E-SMC through two experiments. First, to establish their maximum potential, we apply them to PC-Sampler (Huang et al., 2025a) on `LLaDA` models. This combination sets a new

Table 2: **Accuracy (%) of our path search algorithms on `LLaDA` models across five reasoning and planning benchmarks.** When applied to the strong PC-Sampler baseline, both E-BoN and E-SMC consistently improve performance. Gains over the baseline are highlighted in red.

| Strategies | GSM8K | MATH500 | GPQA | Countdown | Sudoku | Avg.↑ |
|---|---|---|---|---|---|---|
| **LLaDA-Instruct-8B** | | | | | | |
| Uniform | 48.8 | 15.0 | 29.0 | 14.4 | 2.2 | 21.9 |
| Confidence | 6.8 | 3.4 | 27.9 | 34.0 | 23.8 | 19.2 |
| Entropy | 2.2 | 3.8 | 28.4 | 33.8 | 1.6 | 14.0 |
| Margin | 11.1 | 1.8 | 28.4 | 33.9 | 26.6 | 20.4 |
| P2 | 11.6 | 3.4 | 28.2 | 34.2 | 24.0 | 20.3 |
| EB-Sampler | 1.6 | 3.6 | **29.9** | 34.1 | 24.2 | 18.7 |
| Semi-AR | 77.9 | 27.6 | 27.7 | 32.6 | 0.0 | 33.2 |
| Fast-dLLM | 78.2 | 28.4 | 28.6 | 11.4 | 0.3 | 29.4 |
| PC-Sampler | **79.3** | **34.0** | 28.6 | **36.3** | 27.6 | **41.2** |
| *w/* E-BoN | 0.5 ↑ | 1.0 ↑ | 0.2 ↑ | 5.9 ↑ | 0.6 ↑ | 1.6 ↑ |
| *w/* E-SMC | 1.9 ↑ | 1.6 ↑ | 0.3 ↑ | 4.1 ↑ | 1.6 ↑ | 1.9 ↑ |
| **LLaDA-1.5-8B** | | | | | | |
| Uniform | 52.7 | 20.0 | 28.1 | 15.8 | 3.4 | 24.0 |
| Confidence | 19.2 | 5.4 | **29.0** | 33.8 | 24.8 | 22.4 |
| Entropy | 12.1 | 5.0 | 28.8 | 34.7 | 0.2 | 16.2 |
| Margin | 27.9 | 6.4 | 28.6 | 31.8 | **33.6** | 25.7 |
| P2 | 25.4 | 5.6 | 28.8 | 33.6 | 27.8 | 24.2 |
| EB-Sampler | 12.3 | 4.8 | 28.6 | 34.6 | 1.6 | 16.4 |
| Semi-AR | 80.7 | 34.2 | 26.1 | 32.4 | 0.0 | 34.7 |
| Fast-dLLM | 80.8 | 31.2 | 27.9 | 32.9 | 0.4 | 34.6 |
| PC-Sampler | **82.2** | **37.4** | 28.8 | **35.0** | 33.4 | **43.4** |
| *w/* E-BoN | 0.7 ↑ | 0.8 ↑ | 0.1 ↑ | 5.2 ↑ | 0.8 ↑ | 1.5 ↑ |
| *w/* E-SMC | 1.0 ↑ | 1.2 ↑ | 0.2 ↑ | 4.3 ↑ | 0.2 ↑ | 1.4 ↑ |

state-of-the-art, achieving substantial accuracy gains across five challenging benchmarks as shown in Table 2. Second, to validate their broad applicability, we integrate our methods with five different baseline samplers on `Open-dCoder` model. As shown by the average accuracy improvements in Figure 4, both algorithms consistently boost every tested sampler, confirming they act as versatile enhancers for a wide array of decoding strategies.

**Results: path-level optimization is effective for complex reasoning and planning.** The gains are most pronounced on benchmarks requiring multi-step reasoning and planning. For example, on `LLaDA-Instruct-8B`, E-SMC improves GSM8K accuracy from 79.3% to 81.2% (+1.9%) and the Countdown planning task from 36.3% to 40.4% (+4.1%). These results demonstrate that success on such tasks hinges on a global perspective of the entire solution. Our path search algorithms provide this view, leveraging Denoising Entropy to assess the coherence of the entire generation path, not just a single step.

### 4.4 OTHER RESULTS AND ABLATION STUDIES

**Path, $H_{\mathrm{DE}}$, and accuracy.** We validate Denoising Entropy as a proxy for task performance on Sudoku, a benchmark sensitive to decoding order. We use the PC-Sampler hyperparameter $\lambda$ to control decoding sequentiality, thus generating a spectrum of distinct paths. Smaller $\lambda$ encourage more random decoding orders, which benefit Sudoku solving, while larger $\lambda$ values lead to more sequential decoding, harming performance. Figure 5 reveals a negative correlation between $H_{\mathrm{DE}}$ and final accuracy, confirming that higher entropy signals a lower-quality generation path.

**Budget-Efficiency.** Under the same computational budget (using `LLaDA-Instruct-8B` with 5 particles), our entropy-guided methods outperform Majority Vote. Specifically, on the Sudoku and Countdown task, Majority Vote improves accuracy by (-0.8%, +1.0%), while E-BoN and E-SMC achieve gains of (+0.6%, +5.9%) and (+1.6%, +4.1%), respectively. These results indicate that the

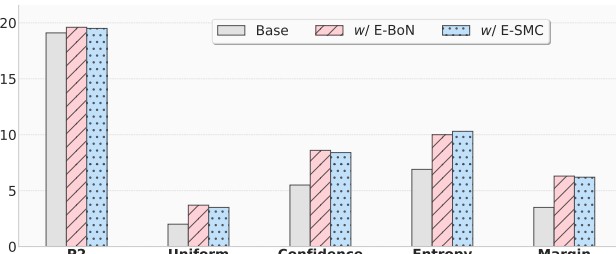 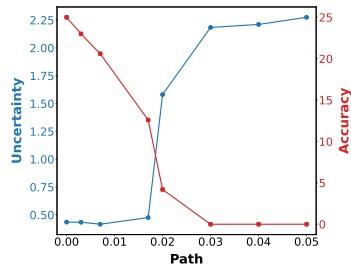

Figure 4: Average accuracy on code benchmarks. E-BON and E-SMC consistently enhance performance of various baseline decoding strategies, demonstrating their broad applicability.

Figure 5: Relationship between decoding path, path uncertainty, and accuracy on Sudoku benchmark.

entropy-guided approach utilizes the sampling budget more effectively than standard ensembling, suggesting greater potential for scaling.

## 5 RELATED WORK

**Masked Diffusion Models.** Recent years have seen the extension of diffusion models, originally successful in continuous domains (Ho et al., 2020; Song et al., 2020; Dhariwal & Nichol, 2021; Rombach et al., 2022), to discrete generation (Austin et al., 2021a). A key branch is Masked Diffusion Models, which was formalized in D3PM (Austin et al., 2021a) and explored in text and sequence generation in DiffusionBERT (He et al., 2023). The field matured with substantial theoretical progress simplifying the training objective, making the paradigm more stable and efficient (Zheng et al., 2023; Sahoo et al., 2024; Shi et al., 2024; Ou et al., 2024; Gat et al., 2024). This progress has culminated in the development of large-scale MDMs (Nie et al., 2025; Zhu et al., 2025; Ye et al., 2025b; Gong et al., 2025a;b) that achieve performance competitive with autoregressive counterparts, establishing MDMs as a viable and powerful paradigm for language generation.

**Decoding Strategies for Language Models.** Decoding strategy is critical for language models. Unlike ARMs, which only select the next token (Graves, 2012), MDMs face a more complex two-dimensional problem: choosing both which position to unmask and what token to generate. Uncertainty-based greedy approaches are most common, where the next position is selected based on local confidence signals like maximum probability (Chang et al., 2022), minimum entropy (Koh et al., 2024; Ben-Hamu et al., 2025), or largest margin (Kim et al., 2025), often building upon a simple random-selection baseline (Austin et al., 2021a). Other approaches impose structure through a semi-autoregressive block-wise order (Han et al., 2023; Nie et al., 2025; Arriola et al., 2025), combining positional bias with content-aware confidence scores (Huang et al., 2025a), or explicitly training a planner (Huang et al., 2025b) to achieve more global control. Some methods improve flexibility by allowing already-decoded tokens to be remasked (Wang et al., 2025; Peng et al., 2025a).

## 6 CONCLUSION AND DISCUSSION

Generation uncertainty has remained a critical but unquantified aspect for MDMs. Our work is the first to address this gap, introducing Denoising Entropy as a metric designed to explicitly quantify the path-level uncertainty of MDM outputs. We also proposed E-BON and E-SMC, two algorithms that leverage Denoising Entropy to actively guide the generation process toward paths with lower uncertainty. Our experiments demonstrate that these methods significantly improve both the general quality of MDM outputs and performance on challenging benchmarks. Beyond the specific algorithms proposed, we believe Denoising Entropy serves as a foundational tool for uncertainty quantification in MDMs, opening directions such as developing more calibrated decoding strategies and providing internal reward signals for reinforcement learning.

**Ethics statement**  We confirm that this research does not raise any ethical concerns. Our work presents a methodological contribution that does not involve human subjects, sensitive data, or potential societal harms.

**Reproducibility Statement**  To facilitate the reproducibility of our results, we have included detailed experimental configurations and hyperparameters in Appendix E.2. We also provide source code and scripts in the supplementary materials to replicate the empirical results. Our implementation is designed to be compatible with existing open-source codebases, which we believe will ease the process of replication and future research.

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

CONTENTS

## A  E-SMC IMPLEMENTATION DETAILS

This appendix provides specific formulas for the **Evaluation** step of the E-SMC algorithm, which were abstracted in the main text for clarity.

**Reward Definition.**   The raw State Entropy, $h_{\text{DE}}(\mathbf{z}) \in [0, \log_2 V]$ where $V$ is the vocabulary size, is first converted into a normalized reward signal $r(\mathbf{z}) \in [0, 1]$. This is done by inverting and scaling the entropy:

$$r(\mathbf{z}) \triangleq \frac{\log_2 V - h_{\text{DE}}(\mathbf{z})}{\log_2 V} = 1 - \frac{h_{\text{DE}}(\mathbf{z})}{\log_2 V}. \tag{12}$$

This formulation ensures that a state with minimum entropy (0) receives the maximum reward (1), and a state with maximum entropy ($\log_2 V$) receives the minimum reward (0).

**Potential Function.**   The reward $r(\mathbf{z})$ is then used to compute the potential score $G(\mathbf{z})$ for each particle, which determines its fitness for resampling. We employ a Gibbs potential function, controlled by a temperature parameter $\lambda > 0$:

$$G(\mathbf{z}) \triangleq \exp(\lambda \cdot r(\mathbf{z})). \tag{13}$$

The parameter $\lambda$ controls the selection pressure. A higher $\lambda$ value more strongly favors particles with high rewards (low entropy), leading to a more aggressive pruning of undesirable trajectories.

**Resampling Probabilities.**   The probability $\pi^{(k)}$ of selecting particle $\mathbf{z}_{t_{i-1}}^{(k)}$ during the resampling step is given by its normalized potential score:

$$\pi^{(k)} = \frac{G(\mathbf{z}_{t_{i-1}}^{(k)})}{\sum_{j=1}^{K} G(\mathbf{z}_{t_{i-1}}^{(j)})} = \frac{\exp(\lambda \cdot r(\mathbf{z}_{t_{i-1}}^{(k)}))}{\sum_{j=1}^{K} \exp(\lambda \cdot r(\mathbf{z}_{t_{i-1}}^{(j)}))}. \tag{14}$$

This is equivalent to applying a softmax function to the scaled rewards of all particles in the population. The new population is then formed by drawing $K$ samples with replacement from the current population according to these probabilities.

## B  THEORETICAL ANALYSIS

### B.1  PROOF OF PROPOSITION 1

As the sample mean of marginal entropy over the masked set $\mathcal{M}_t$, $h_{\text{DE}}(\mathbf{z}_t)$ is the expected entropy for the single position $\ell$ chosen uniformly from $\mathcal{M}_t$:

$$h_{\text{DE}}(\mathbf{z}_t) \triangleq \frac{1}{|\mathcal{M}_t|} \sum_{\ell \in \mathcal{M}_t} H\left(p_\theta(X_0^\ell|\mathbf{z}_t, t)\right) = \mathbb{E}_{\ell \sim \text{Unif}(\mathcal{M}_t)}\left[H\left(p_\theta(X_0^\ell|\mathbf{z}_t, t)\right)\right].$$

This establishes $h_{\text{DE}}$ as a measure of average uncertainty over masked tokens.

Our proof relies on the subadditivity property of Shannon entropy. We state it here as a lemma.

> **Lemma 1 (Subadditivity of Conditional Entropy) .**   *For any set of random variables $\{Y_1, \ldots, Y_n\}$ and a conditioning variable $Z$, the joint conditional entropy is bounded by the sum of the individual conditional entropy:*
>
> $$H(Y_1, \ldots, Y_n|Z) \leq \sum_{i=1}^{n} H(Y_i|Z).$$
>
> *Equality holds if and only if the variables $\{Y_i\}$ are conditionally independent given $Z$.*

We now restate and prove Proposition 1.

**Proposition 1.** *Oracle State Uncertainty $H_{oracle}(\mathbf{z}_t)$ is upper-bounded by the sum of the marginal entropies, which is directly proportional to State Entropy $h_{DE}(\mathbf{z}_t)$:*

$$H_{\text{oracle}}(\mathbf{z}_t) \leq |\mathcal{M}_t| \cdot h_{\text{DE}}(\mathbf{z}_t).$$

*Proof.* By definition, $H_{\text{oracle}}(\mathbf{z}_t)$ is the conditional joint entropy $H(\mathbf{X}_{\mathcal{M}_t}|\mathbf{z}_t)$. We can derive the bound as follows:

$$
\begin{aligned}
H_{\text{oracle}}(\mathbf{z}_t) &= H(\mathbf{X}_{\mathcal{M}_t}|\mathbf{z}_t) && \text{by Definition 1} \\
&\leq \sum_{\ell \in \mathcal{M}_t} H(X_0^\ell|\mathbf{z}_t) && \text{by Lemma 1} \\
&= \sum_{\ell \in \mathcal{M}_t} H\left(p_\theta(X_0^\ell|\mathbf{z}_t, t)\right) && \text{by definition of MDM output} \\
&= |\mathcal{M}_t| \cdot \left(\frac{1}{|\mathcal{M}_t|} \sum_{\ell \in \mathcal{M}_t} H\left(p_\theta(X_0^\ell|\mathbf{z}_t, t)\right)\right) \\
&= |\mathcal{M}_t| \cdot h_{\text{DE}}(\mathbf{z}_t). && \text{by Definition 2}
\end{aligned}
$$

The inequality in the second step is strict if the random variables $\{X_0^\ell\}_{\ell \in \mathcal{M}_t}$ are not conditionally independent given the latent state $\mathbf{z}_t$. In the context of natural language, where token occurrences are highly correlated, this condition for strict inequality is almost always met. $\square$

### B.2 PROOF AND COROLLARY OF PROPOSITION 2

Here, we provide the detailed proof for Proposition 2 and discuss the correlation between negative evidence lower bound $\mathcal{L}$ and $H_{\text{DE}}$.

The proof relies on the decomposition of cross-entropy loss:

> **Lemma 2 (Decomposition of Cross-Entropy Loss)** . *Let $q(X)$ be the true data distribution and $p_\theta(X)$ be the model's estimation distribution. For any random variable $X$, the cross-entropy loss can be decomposed as:*
>
> $$\mathbb{E}_{X_0^\ell \sim q(\cdot|\mathbf{z}_t)}[-\log p_\theta(X_0^\ell|\mathbf{z}_t, t)] = H(q) + D_{KL}(q \parallel p_\theta(X_0^\ell|\mathbf{z}_t, t)),$$
>
> *where $q = q(X_0^\ell|\mathbf{z}_t)$ denotes the true posterior distribution, $H(\cdot)$ is the Shannon entropy and $D_{KL}(\cdot \parallel \cdot)$ is the Kullback-Leibler divergence.*

The proposition states that with the assumption of $\epsilon$-accurate model, the gap between $h_{\text{DE}}$ and training loss can be upper-bounded:

$$\frac{1}{|\mathcal{M}_t|} \sum_{\ell \in \mathcal{M}_t} \left(-\log p_{\boldsymbol{\theta}}(x_0^\ell|\mathbf{z}_t, t)\right) - h_{\text{DE}}(\mathbf{z}_t) \leq \epsilon + \sqrt{\frac{\epsilon}{2}} log K \,. \tag{15}$$

*Proof.* For an $\epsilon$-accurate model, the KL divergence between the posterior distribution $q(x_0^\ell|\mathbf{z}_t)$ and the estimation distribution $p_\theta(x_0^\ell|z_t, t)$, which is less than $D_{\text{KL}}\left(q(\mathbf{z}_{t_{i-1}}|\mathbf{z}_{t_i}) \parallel p_\theta(\mathbf{z}_{t_{i-1}}|\mathbf{z}_{t_i})\right)$, is upper-bounded by $\epsilon$:

$$D_{\text{KL}}(q(x_0^\ell|\mathbf{z}_t) \parallel p_\theta(x_0^\ell|\mathbf{z}_t, t)) \leq \epsilon \,. \tag{16}$$

With the upper bound as above, we turn to bound the gap between entropy $H(p_\theta(x_0^\ell|\mathbf{z}_t, t))$ and $H(q(x_0^\ell|\mathbf{z}_t))$. We use $q, p_\theta$ for simplification.

Let $\delta(q, p_\theta) \triangleq \frac{1}{2} \sum_{x \in \mathcal{V}} |q(x) - p_\theta(x)|$ be the total variation distance on the size-$K$ vocabulary set $\mathcal{V}$. Since $D_{\text{KL}}(q \parallel p_\theta) \leq \epsilon$, we obtain $\delta(q, p_\theta) \leq \sqrt{\frac{\epsilon}{2}}$ based on Pinsker's Inequality (Csiszár & Körner, 2011).

Using Audenaert-Fannes Inequality, we obtain the bound of shannon entropy gap:

$$|H(q) - H(p_\theta)| \leq \delta \log(K-1) + h(\delta), \tag{17}$$

where $h(\delta)$ is binary entropy function $h(\delta) = -\delta log \delta - (1-\delta) log(1-\delta)$. Since $h(\delta)$ (bounded with $\delta \leq \sqrt{\frac{\epsilon}{2}}$) is small term that approches 0 when $\epsilon$ approaches 0, we omit this term in the following inequality. Thus, we have $|H(q) - H(p_\theta)| \leq \delta \log(K)$.

Putting the above inequalities together, we obtain the final result:

$$\mathbb{E}_{X_0^\ell \sim q(\cdot|\mathbf{z}_t)}[-\log p_\theta(X_0^\ell|\mathbf{z}_t, t)] \leq h_{\text{DE}}(\mathbf{z}_t) + \epsilon + \sqrt{\frac{\epsilon}{2}} log K. \tag{18}$$

By the law of large numbers, the average loss over masked positions approximates the expectation:

$$\frac{1}{|\mathcal{M}_t|} \sum_{\ell \in \mathcal{M}_t} \left( -\log p_{\boldsymbol{\theta}}(x_0^\ell|\mathbf{z}_t, t) \right) \approx \mathbb{E}_{X_0^\ell \sim q(\cdot|\mathbf{z}_t)}[-\log p_\theta(X_0^\ell|\mathbf{z}_t, t)],$$

$$\frac{1}{|\mathcal{M}_t|} \sum_{\ell \in \mathcal{M}_t} \left( -\log p_{\boldsymbol{\theta}}(x_0^\ell|\mathbf{z}_t, t) \right) - h_{\text{DE}}(\mathbf{z}_t) \leq \epsilon + \sqrt{\frac{\epsilon}{2}} log K.$$

$$\square$$

**Justification for the Correlation between $\mathcal{L}$ and $H_{\textbf{DE}}$.** The full objective (negative evidence lower bound, NELBO) $\mathcal{L}$ and our metric $H_{\text{DE}}(\tau)$ are strongly correlated, which is direct corollary of Proposition 2.

We begin with the definition of the NELBO objective:

$$\mathcal{L}(\mathbf{x}_0) = \int_0^1 w(t) \left[ \sum_{\ell \in \mathcal{M}_t} -\log p_\theta(x_0^\ell|\mathbf{z}_t, t) \right] dt, \tag{19}$$

where the weighting function is $w(t) = |\frac{d\alpha_t}{dt}|\frac{1}{1-\alpha_t}$. Assume that $\epsilon \to 0$, we take the approximation by $h_{\text{DE}}$ from Proposition 2:

$$\sum_{\ell \in \mathcal{M}_t} -\log p_\theta(x_0^\ell|\mathbf{z}_t, t) \approx |\mathcal{M}_t| \cdot h_{\text{DE}}(\mathbf{z}_t). \tag{20}$$

Substituting this back into the $\mathcal{L}(\mathbf{x}_0)$:

$$\mathcal{L} \approx \int_0^1 w(t) \cdot |\mathcal{M}_t| \cdot h_{\text{DE}}(\mathbf{z}_t) dt. \tag{21}$$

In expectation, $|\mathcal{M}_t| \approx L(1 - \alpha_t)$, where $L$ is the sequence length. Thus, the term $w(t) \cdot |\mathcal{M}_t| \approx L \cdot |\frac{d\alpha_t}{dt}|$ is a strictly positive weighting function of time $t$, which we denote as $w'(t)$. Then the objective is approximated as follows:

$$\mathcal{L} \approx \int_0^1 w'(t) \cdot h_{\text{DE}}(\mathbf{z}_t) dt. \tag{22}$$

In parallel, our metric is the unweighted integral:

$$H_{\text{DE}}(\tau) = \int_0^1 h_{\text{DE}}(\mathbf{z}_t) dt. \tag{23}$$

Since both $\mathcal{L}$ and $H_{\text{DE}}(\tau)$ are integrals of the same underlying function, $h_{\text{DE}}(\mathbf{z}_t)$, with one being uniformly weighted and the other weighted by a positive function $w'(t)$, a strong positive correlation between their values is mathematically expected. This provides rigorous justification for the remark.

## C  THEORETICAL ANALYSIS OF DECODING ALGORITHMS

In this section, we provide a rigorous theoretical justification for minimizing Path Uncertainty as a proxy for improving generation quality. We establish a direct link between our proposed metric and the divergence from the true data distribution.

### C.1  DISTRIBUTIONS OVER PATHS

To rigorously compare the uncertainty of generated samples against real data, we first align their metrics. While $H_{\text{DE}}(\tau)$ is defined over a decoding path $\tau = (\mathbf{z}_{t_N}, \ldots, \mathbf{z}_{t_0})$, real data only consists of static sample sequence $\mathbf{x}_0$. We bridge this gap by defining the **True Path Distribution** via the forward diffusion process.

**Definition 5 (Path Distributions) .** *We consider two distributions over the space of paths:*

- **Model Distribution ($\widehat{\mathcal{T}}$):** *The distribution of paths generated by the model's reverse process, starting from Gaussian noise:*

$$\widehat{\Pr}(\tau) = p(\mathbf{z}_{t_N}) \prod_{i=N}^{1} p_\theta(\mathbf{z}_{t_{i-1}}|\mathbf{z}_{t_i}) , \tag{24}$$

*where the starting state is sampled from the prior $\mathbf{z}_{t_N} \sim p(\cdot)$.*
- **Reference (True) Distribution ($\mathcal{T}^*$):** *The distribution of paths induced by applying the forward corruption process $q(\mathbf{z}_t|\mathbf{x}_0)$ to ground-truth data $\mathbf{x}_0 \sim \mathcal{D}_{real}$:*

$$\Pr(\tau) = q(\mathbf{x}_0) \prod_{i=1}^{N} q(\mathbf{z}_{t_i}|\mathbf{z}_{t_{i-1}}, \mathbf{x}_0) , \tag{25}$$

*where $\mathbf{x}_0 \sim q(\cdot)$ is sampled from the real dataset $\mathcal{D}_{real}$.*

Here, $\Pr(\tau)$ represents the distributions of the "ideal" paths that stay perfectly on the manifold of the forward process. For a well-trained model, evaluating $H_{\mathrm{DE}}$ on the reversed reference paths should yield low uncertainty, as the states are consistent with the training objective. Conversely, generated paths $\tau \sim \widehat{\Pr}$ that deviate from this manifold (e.g., via hallucinations) are expected to induce higher model uncertainty.

The discrepancy between the generated paths and the true data manifold originates from imperfect approximation at each denoising step. Assume that MEM is $\epsilon$-accurate. Although $\epsilon$ is small locally, the generation process involves a sequence of $N$ transitions. For the full path distributions defined as $\Pr(\tau)$ (Reference) and $\widehat{\Pr}(\tau)$ (Model), the chain rule of KL divergence implies that errors accumulate additively:

$$D_{\mathrm{KL}}(\Pr(\tau)\|\widehat{\Pr}(\tau)) = \sum_{i=N}^{1} \mathbb{E}_{\mathbf{z}_{t_i} \sim q} \left[ D_{\mathrm{KL}}(q(\cdot|\mathbf{z}_{t_i})\|p_\theta(\cdot|\mathbf{z}_{t_i})) \right] \leq N \cdot \epsilon . \tag{26}$$

Equation 26 establishes that even a well-trained model ($\epsilon \to 0$) can exhibit a non-negligible path divergence $N\epsilon$ due to the cumulative nature of the generative process. This macroscopic divergence is the root cause of the "drift" phenomenon.

## C.2 ENTROPY DRIFT AND THE DIVERGENCE BOUNDS

### C.2.1 PROOF OF PROPOSITION 3

We formally quantify the entropy discrepancy as **Entropy Drift**. Let $\mu_{\Pr} = \mathbb{E}_{\tau \sim \Pr}[H_{\mathrm{DE}}(\tau)]$ be the expected Path Entropy evaluated on reference paths (representing the model's uncertainty when guided by ground truth), and let $\mu_{\widehat{\Pr}} = \mathbb{E}_{\tau \sim \widehat{\Pr}}[H_{\mathrm{DE}}(\tau)]$ be the expected Path Entropy of the model's autonomous generations.

*Proof.* We now show that the gap between the two Path Entropy quantities lower-bounds the divergence between the generated distribution and the true distribution.

**Lemma 3 (Pinsker's Inequality) .** *Let $P$ and $Q$ be two distributions over the path space. For any bounded measurable functional $f(\tau)$ such that $|f(\tau)| \leq B$, the difference in expectations is bounded by the KL divergence:*

$$|\mathbb{E}_{\tau \sim P}[f(\tau)] - \mathbb{E}_{\tau \sim Q}[f(\tau)]| \leq B\sqrt{2D_{\mathrm{KL}}(P\|Q)} . \tag{27}$$

Applying Lemma 3 with $f(\tau) := H_{\mathrm{DE}}(\tau)$, we obtain the following bound:

$$D_{\mathrm{KL}}(\Pr\|\widehat{\Pr}) \geq \frac{1}{2B^2} \left( \mu_{\widehat{\Pr}} - \mu_{\Pr} \right)^2 .$$

$\square$

**Remark.** Equation ablove establishes a necessary condition for high-fidelity generation. Ideally, we expect the generated distribution to match the reference distribution ($D_{\mathrm{KL}} \to 0$), but Theorem 3 implies that it is impossible unless the entropy of the generated paths matches the entropy of the reference paths ($\mu_{\widehat{\mathrm{Pr}}} \to \mu_{\mathrm{Pr}}$). Since reference paths $\mathrm{Pr}$ correspond to ground-truth data evolution where the model is typically confident (low $\mu_{\mathrm{Pr}}$), and autonomous generations often drift into high-uncertainty regions (high $\mu_{\widehat{\mathrm{Pr}}}$), minimizing the expected Path Entropy $\mu_{\widehat{\mathrm{Pr}}}$ effectively reduces the gap, implicitly constraining the generative process closer to the true data manifold.

### C.2.2 ENTROPY GAP LOWER BOUNDS DIVERGENCE

Since computing the high-dimensional divergence $D_{\mathrm{KL}}(\mathrm{Pr}\,\|\,\widehat{\mathrm{Pr}})$ directly is intractable, we propose using Path Entropy as a measurable proxy. We formally show that the KL divergence is upper-bounded with $\epsilon$-accurate condition and lower-bounded with the measurable gap in expected path entropy.

---

**Theorem 1 (Bounds of KL Divergence) .** *Let* $\mathrm{Pr}$ *and* $\widehat{\mathrm{Pr}}$ *be the reference and generated distributions over path space;* $\mu_{\mathrm{Pr}} = \mathbb{E}_{\tau \sim \mathrm{Pr}}[H_{DE}(\tau)]$ *and* $\mu_{\widehat{\mathrm{Pr}}} = \mathbb{E}_{\tau \sim \widehat{\mathrm{Pr}}}[H_{DE}(\tau)]$ *be the Path Entropy expectation on the reference and model distributions, respectively.*
*Assume that MDM is* $\epsilon - accurate$ *for the model's reverse kernel* $p_\theta$ *and true transition kernel* $q$, *the KL-divergence of distributions* $\mathrm{Pr}$ *and* $\widehat{\mathrm{Pr}}$ *is upper-bounded as in Eq. 26. By applying Pinsker's Inequality to path functionals, the divergence is lower-bounded by the squared entropy gap:*

$$N \cdot \epsilon \geq D_{\mathrm{KL}}(\mathrm{Pr}\,\|\,\widehat{\mathrm{Pr}}) \geq \frac{1}{2B^2}\left(\mu_{\widehat{\mathrm{Pr}}} - \mu_{\mathrm{Pr}}\right)^2 . \tag{28}$$

---

**Implication:** This inequality chain reveals the full mechanism:

- **Source:** Local approximation errors ($\epsilon$) accumulate over $N$ steps.
- **State:** This accumulation creates a distributional shift $D_{\mathrm{KL}}(\mathrm{Pr}\,\|\,\widehat{\mathrm{Pr}})$.
- **Observation:** This shift forces drift in observable statistics, specifically creating a gap $|\mu_{\widehat{\mathrm{Pr}}} - \mu_{\mathrm{Pr}}|$.

Therefore, if we observe a large entropy gap (Entropy Drift), it implies that the model has deviated significantly from the reference path. Conversely, our algorithms aim to reduce this entropy gap $|\mu_{\widehat{\mathrm{Pr}}} - \mu_{\mathrm{Pr}}| \to 0$, which is a necessary condition for tightening the divergence bound and mitigating the accumulated error.

### C.3 ENTROPY DRIFT IN DECODING ALGORITHMS

Motivated by Theorem 3, our algorithms, E-BoN and E-SMC, aim to bridge the divergence gap by aligning the expected path entropy of the generated distribution $\mu_{\widehat{\mathrm{Pr}}}$ with that of the reference $\mu_{\mathrm{Pr}}$.

**E-BoN.** E-BoN approximates a truncated distribution $\widehat{\mathrm{Pr}}_{\text{E-BoN}}$ by rejecting paths with high $H_{\mathrm{DE}}$. A potential concern is that the algorithm might retain path samples with extremely low uncertainty (e.g., repetition). For a well-designed model $\widehat{\mathrm{Pr}}$, the sampling probability of such paths is low. The model tends to sample problematic sequences with high entropy or sequences with normal entropy; then the finite candidate path set $S = \{\tau_1, ... \tau_N\}$ is unlikely to contain path samples with extremely low entropy. E-BoN reduces the expectation $\mathbb{E}_{\tau \sim \widehat{\mathrm{Pr}}_{\text{E-BoN}}}[H_{\mathrm{DE}}(\tau)]$. Considering that $S$ consists solely of samples from the region where $\mu_{\widehat{\mathrm{Pr}}} \leq \mu_{\mathrm{Pr}}$, the selection process brings it closer to the reference baseline $\mu_{\mathrm{Pr}}$. By Theorem 3, bridging this path entropy gap is a prerequisite for reducing the distributional divergence.

**E-SMC.** E-SMC modifies the transition kernel by re-weighting particles with $w \propto \exp(-\lambda h_{\mathrm{DE}})$ at fixed intervals of $\Delta_{i_r}$ steps, which can be viewed as constructing a calibrated distribution $\widehat{\mathrm{Pr}}_{\text{E-SMC}}$ that penalizes transitions into high-entropy states. By actively avoiding regions where the model is uncertain (which are characteristic of $\widehat{\mathrm{Pr}}$ but rare in $\mathrm{Pr}$), E-SMC promotes that the accumulated Path Entropy remains low. A similar potential concern is that when the entropy of $\mu_{\widehat{\mathrm{Pr}}}$ is lower than the reference $\mu_{\mathrm{Pr}}$, reducing the entropy $\mu_{\widehat{\mathrm{Pr}}}$ would increase the gap. There we can set the temperature $\lambda$ near the optimal one $\lambda^*$, which enables $\mu_{\widehat{\mathrm{Pr}}} = \mu_{\mathrm{Pr}}$ and the entropy gap is zero. The existence of $\lambda^*$ can be verified as follows:

**Theorem 2 (Existence and Uniqueness of Optimal Temperature) .** *Let $\mu_{\widehat{\mathrm{Pr}}}(\lambda)$ denote the expected path entropy of the generative process under the* E-SMC *algorithm with temperature parameter $\lambda \in [0, \infty)$. There exists a unique optimal temperature $\lambda^* \in (0, \infty)$ such that the expected generated entropy matches the reference entropy:*

$$\mu_{\widehat{\mathrm{Pr}}}(\lambda^*) = \mu_{\mathrm{Pr}} \,. \tag{29}$$

*Proof.* Consider a set of $M$ particles with state entropies $\{H_1, \ldots, H_M\}$ before evaluation, the $m$-th particle is retained with probability $\pi_m(\lambda) = \frac{\exp(-\lambda H_m)}{\sum_{j=1}^{M} \exp(-\lambda H_j)}$ in the E-SMC resampling step. The expected entropy after resampling is given by the function $\tilde{H} : [0, \infty) \to \mathbb{R}$:

$$\tilde{H}(\lambda) = \sum_{m=1}^{M} \pi_m(\lambda) H_m = \frac{\sum_{m=1}^{M} H_m \exp(-\lambda H_m)}{\sum_{j=1}^{M} \exp(-\lambda H_j)} \,. \tag{30}$$

**Continuity.** Since the denominator $\sum_{j=1}^{M} \exp(-\lambda H_j)$ is a sum of strictly positive terms for all $\lambda \in \mathbb{R}$, $\tilde{H}(\lambda)$ is continuous and differentiable on $[0, \infty)$.

**Monotonicity.** Since $\frac{d}{d\lambda} e^{-\lambda H} = -H e^{-\lambda H}$, we obtain:

$$\frac{d}{d\lambda} \tilde{H}(\lambda) = -\left( \sum_{m=1}^{M} \pi_m(\lambda) H_m^2 - \left( \sum_{m=1}^{M} \pi_m(\lambda) H_m \right)^2 \right) = -\mathrm{Var}\pi(\lambda)(H). \tag{31}$$

Since the variance $\mathrm{Var}\pi(\lambda)(H)$ is non-negative, the derivative $\frac{d}{d\lambda}$ is non-positive. Assuming the particle entropies $\{H_m\}$ are not all identical, the variance is strictly positive, implying $\tilde{H}(\lambda)$ is strictly decreasing with respect to $\lambda$.

- **Case $\lambda = 0$:** The weights become uniform with $\pi_m(0) = 1/M$. The output distribution is consistent with the original model. Given the Entropy Drift phenomenon, we have $\mu_{\widehat{\mathrm{Pr}}} > \mu_{\mathrm{Pr}}$ when $\lambda = 0$.
- **Case $\lambda \to \infty$:** The retaining probability of resampling steps concentrates on the particle with the minimum entropy, that is,

$$\lim_{\lambda \to \infty} \pi_m(\lambda) = \begin{cases} 1, & \text{if } H_m = \min_i H_i \,, \\ 0, & \text{otherwise} \,. \end{cases}$$

This corresponds to Model Collapse, where $\lim_{\lambda \to \infty} \mu_{\widehat{Pr}}(\lambda) < \mu_{\mathrm{Pr}}$.

The function $\mu_{\widehat{\mathrm{Pr}}}(\lambda)$ is continuous and strictly monotonically decreasing on the domain $[0, \infty)$. The reference entropy $\mu_{\mathrm{Pr}}$ lies strictly between the boundary values $\lim_{\lambda \to \infty} \mu_{\widehat{\mathrm{Pr}}}(\lambda)$ and $\mu_{\widehat{\mathrm{Pr}}}(0)$. By the **Intermediate Value Theorem**, there exists a $\lambda^*$ such that $\mu_{\widehat{\mathrm{Pr}}}(\lambda^*) = \mu_{\mathrm{Pr}}$. Furthermore, due to the strict monotonicity of the function, this $\lambda^*$ is unique. $\qquad\square$

**Remark.** While Theorem 2 establishes $\lambda$ as a primary control for entropy, the resampling interval $\Delta i_r$ also governs the final uncertainty. Since the autonomous generative process inherently exhibits *Entropy Drift*, uncertainty accumulates between resampling steps. Consequently, a larger $\Delta i_r$ allows greater accumulation of drift before resampling occurs, leading to a higher expectation of entropy. Thus, $\lambda$ (resampling strength) and $\Delta i_r$ (resampling frequency) jointly determine the accumulated entropy of the generation.

## D THEORETICAL PROPERTIES OF STATE ENTROPY

We present theoretical analysis to verify the validity of the state entropy $h_{\mathrm{DE}}(\mathbf{z}_t)$. We begin by verifying its asymptotic behavior at the diffusion process boundaries. Then, we establish its monotonicity with respect to context information.

> **Theorem 3 (Asymptotic Behavior of state Entropy) .** *Let $\hat{\mathbf{p}}_\theta(\cdot|\mathbf{z}_t, t)$ be a denoising model that approximates the true posterior distribution $q(\mathbf{z}_t|x_0)$. Assume the following limits exist, then the expected state entropy $\mathbb{E}[h_{DE}(\mathbf{z}_t)]$ satisfies:*
>
> - *$\textbf{Fully-Noised State}$ ($t \to 1$): As $t \to 1$, $\alpha_t \to 0$ and $\mathbf{z}_1$ becomes independent of the original sequence $\mathbf{x}_0$. The entropy converges to the marginal distribution of tokens in the training data, $p_{data}$*
>
> $$\lim_{t \to 1} \mathbb{E}_{\mathbf{z}_t \sim q(\mathbf{z}_t|\mathbf{x}_0)}[h_{DE}(\mathbf{z}_t)] = H(p_{data}) \,. \tag{32}$$
>
> - *$\textbf{Noise-Free State}$ ($t \to 0$): As $t \to 0$, $\alpha_t \to 1$ and the input $z_0$ is the clean data $x_0$. The model performs reconstruction, and the entropy converges to 0:*
>
> $$\lim_{t \to 0} \mathbb{E}_{\mathbf{z}_t \sim q(\mathbf{z}_t|\mathbf{x}_0)}[h_{DE}(\mathbf{z}_t)] = 0 \,. \tag{33}$$

*Proof.* The proof examines the two temporal boundaries of the diffusion process.

**Asymptotic behavior as $t \to 1$.** As $t \to 1$, the noise schedule satisfies $\lim_{t \to 1} \alpha_t = 0$. The forward process marginal, $q(z_t|x_0) = \text{Cat}(z_t; \alpha_t x_0 + (1 - \alpha_t)\mathbf{m})$, converges to a categorical distribution concentrated entirely on the mask token:

$$\lim_{t \to 1} q(\mathbf{z}_t|\mathbf{x}_0) = \text{Cat}(\mathbf{z}_t; \mathbf{m}) \,, \tag{34}$$

where $\text{Cat}(\cdot; \boldsymbol{\pi})$ denotes the categorical distribution with probability vector $\boldsymbol{\pi}$, and $\mathbf{m}$ is the one-hot vector representing the [MASK] token. Consequently, $\mathbf{z}_1 = [\mathbf{m}, \ldots, \mathbf{m}]$ is deterministic and independent of $\mathbf{x}_0$, with all positions masked ($\mathcal{M}_1 = \{1, \ldots, L\}$). For an optimally trained denoising model, the prediction at every masked position converges to the marginal data distribution:

$$\forall \ell \in \mathcal{M}_1, \quad \lim_{t \to 1} \mathbf{p}_\theta^\ell(\mathbf{z}_t, t) = p_{\text{data}} \,. \tag{35}$$

The state entropy therefore satisfies:

$$h_{\text{DE}}(\mathbf{z}_1) = \frac{1}{L} \sum_{\ell=1}^{L} H(p_{\text{data}}) = H(p_{\text{data}}) \,. \tag{36}$$

As $z_1$ is a deterministic state, the expectation is equal to the value itself.

**Asymptotic behavior as $t \to 0$.** In the opposite limit $t \to 1$, $\alpha_t \to 1$. The forward process marginal converges to the clean data:

$$\lim_{t \to 0} q(\mathbf{z}_t|\mathbf{x}_0) = Cat(\mathbf{z}_t, \mathbf{x}_0) \,, \implies z_0 = x \,. \tag{37}$$

In this regime, the model has access to nearly the entire ground-truth context. For any masked position $\ell \in \mathcal{M}_t$ where $t$ is small, an optimal model's prediction $p_\theta^\ell(x_0|z_t, t)$ is conditioned on a context that is almost entirely the ground truth, $\{x_0^j \mid j \notin \mathcal{M}_t\}$. Formally, the predictive distribution converges to a point mass on the true token:

$$\lim_{t \to 0} \hat{\mathbf{p}}_\theta^\ell(\mathbf{z}_t, t) = \delta_{k, x_0^\ell} \,. \tag{38}$$

The Shannon entropy of a Dirac delta distribution is zero:

$$\lim_{t \to 0} H\left(\hat{x}_\theta^\ell(z_t, t)\right) = H(\delta_{k, x_0^\ell}) = 0 \,. \tag{39}$$

As this holds for every masked position, the average entropy $h_{\text{DE}}(\mathbf{z}_t)$ converges to zero. The convergence of the expectation follows from the deterministic nature of the limit and the boundedness of the entropy function.

This completes the verification of both asymptotic behaviors. $\qquad\square$

The asymptotic analysis confirms that $h_{\text{DE}}$ behaves intuitively at process boundaries. We now establish its monotonicity with respect to context information.

**Lemma 4 (Concavity of Entropy) .** *The Shannon entropy $H(p)$ is a concave function on the probability simplex $\Delta^{K-1}$.*

*Proof.* The entropy function $H(p) = -\sum_{i=1}^{K} p_i \log p_i$ is a sum of terms $f(p_i) = -p_i \log p_i$. Since $f''(p_i) = -1/p_i < 0$ for $p_i > 0$, each $f(p_i)$ is strictly concave. The sum of concave functions remains concave. $\square$

We now turn to investigating the context-sensitivity property of the state entropy. Let $\mathcal{O}_t \subseteq \{1, \ldots, L\}$ denote the set of indices corresponding to unmasked tokens a time $t$.

**Theorem 4 (Context-Sensitivity of State Entropy) .** *Let $\mathbf{z}_t$ and $\mathbf{z}'_t$ be latent variables with observed(unmasked) position sets $\mathcal{O}_t$ and $\mathcal{O}'_t$ respectively, where $\mathcal{O}_t \subset \mathcal{O}'_t$. For any masked position $\ell \notin \mathcal{O}'_t$, revealing additional context does not increase the prediction entropy:*

$$\mathbb{E}[H(\hat{\mathbf{p}}_\theta^\ell(\mathbf{z}'_t, t))] \leq \mathbb{E}[H(\hat{\mathbf{p}}_\theta^\ell(\mathbf{z}_t, t))]. \tag{40}$$

*Proof.* For an optimal denoising model, the prediction $\hat{p}_\theta^\ell(z_t, t)$ approximates the true posterior distribution $p(x_0^\ell | \{x_0^j\}_{j \in \mathcal{O}_t})$, where $\mathcal{O}_t$ is the set of observed indices in $z_t$. Let $X = x_0^\ell$ be the target token at position $\ell$, $Y = \{x_0^j\}_{j \in \mathcal{O}_t}$ be the initial context, and $Z = \{x_0^j\}_{j \in \mathcal{O}'_t \setminus \mathcal{O}_t}$ be the additional context revealed in $z'_t$.

By the information-theoretic property that conditioning reduces entropy:

$$H(X|Y, Z) \leq H(X|Y), \tag{41}$$

the entropy of an optimal model's prediction should converge to the true conditional entropy of the data:

$$\mathbb{E}[H(\hat{\mathbf{p}}_\theta^\ell(\mathbf{z}_t, t))] \rightarrow H(X|Y), \tag{42}$$
$$\mathbb{E}[H(\hat{\mathbf{p}}_\theta^\ell(\mathbf{z}'_t, t))] \rightarrow H(X|Y, Z). \tag{43}$$

Combining these inequalities yields the desired result Equation 40. $\square$

Theorem 4 demonstrates that $h_{\text{DE}}$ properly decreases with increasing context information, validating its use as a measure of uncertainty in masked diffusion models.

# E  EXPERIMENTS

## E.1  DECODING STRATEGIES

Here we provide a detailed description of the decoding strategies we evaluate.

**Uniform (Austin et al., 2021a)**   At each step, this sampler unmasks a fixed number of tokens at positions chosen uniformly at random from the set of currently masked tokens. This is the vanilla sampler in MDMs.

**Confidence (Chang et al., 2022)**   Sampler with confidence score selects tokens for positions where MDM prediction is most confident, defined as the position $\ell$ that maximizes the probability of the most likely token, i.e., $\arg\max_{\ell \in \mathcal{M}_t} \max(\hat{\mathbf{p}}_0^\ell)$.

**Entropy (Ben-Hamu et al., 2025)**   Sampler with entropy selects tokens for positions where the MDM prediction is least ambiguous, defined as the position $\ell$ that minimizes the Shannon entropy of the predicted probability distribution, i.e., $\arg\min_{\ell \in \mathcal{M}_t} H(\hat{\mathbf{p}}_0^\ell)$.

**Margin (Kim et al., 2025)**   Sampler with margin selects tokens for positions with the clearest distinction between the top two candidates, defined as the position $\ell$ that maximizes the margin $p_{(1)}^\ell - p_{(2)}^\ell$, where $p_{(1)}^\ell$ and $p_{(2)}^\ell$ are the highest and second-highest probabilities in $\hat{\mathbf{p}}_0^\ell$.

**EB-Sampler (Ben-Hamu et al., 2025)** Entropy-Bounded sampler is an adaptive method that unmasks a variable number of tokens, $k$, at each step. It selects the $k$ tokens with the lowest entropy, where $k$ is dynamically determined by ensuring the cumulative entropy of the selected tokens remains bounded by a predefined threshold $\gamma$.

**Semi-AR (Nie et al., 2025)** Semi-AutoRegressive sampler partitions the sequence into contiguous blocks. It generates tokens within each block in a parallel, non-autoregressive manner, while the blocks themselves are generated sequentially from left to right.

**Fast-dLLM (Wu et al., 2025)** This sampler accelerates the generation of each block by adaptively unmasking a variable number of tokens at each step. The number of tokens is determined based on whether their prediction confidence exceeds a given threshold, allowing a block to be completed in fewer steps than prescheduled.

**PC-Sampler (Huang et al., 2025a)** Positional-Confidence sampler selects tokens based on a score that combines a confidence measure (derived from the cross-entropy against a background token distribution) with an exponentially decaying positional bias, thus prioritizing tokens that are confidently predicted and appear earlier in the sequence.

**P2 (Peng et al., 2025a)** This is a multi-stage sampler that employs a self-correction mechanism. It first generates a high-proportion draft of the sequence by filling the most confident positions. It then enters an iterative refinement phase, where in each step it identifies the *least* confident generated tokens, re-masks them, and immediately re-predicts their content based on the updated context. This process allows MDM to revise and improve its initial predictions.

### E.2 EXPERIMENTAL CONFIGURATIONS

Here we provide a detailed description of the configurations for each experiment. Table 3 summarizes the hyperparameters used for experiments on `LLaDA` model. Table 4 summarizes the hyperparameters used for experiments on `Open-dCoder` model.

**Note on Selection Temperature Parameter.** `LLaDA` experiments utilize a selection temperature parameter ($T_{\text{sel}} = 0.1$) while `Open-dCoder` experiments do not require this parameter. This difference is because of the generation temperature settings and their impact on path diversity.

In `LLaDA` experiments, generation temperature is set to 0, making the token prediction process deterministic. Under such deterministic conditions, multiple runs would yield identical paths, preventing path exploration for E-BoN and E-SMC algorithms. The selection temperature introduces controlled randomness during the position selection phase of decoding strategies.

Specifically, when selecting the next $k$ positions to unmask based on confidence scores $\mathbf{s} = [s_1, s_2, \ldots, s_n]$, the selection temperature modifies the standard top-$k$ selection through stochastic sampling:

$$\text{Top-}2k \text{ candidates:} \quad \mathbf{s}_{\text{top}} = \text{topk}(\mathbf{s}, 2k) \tag{44}$$

$$\text{Temperature-scaled probabilities:} \quad p_i = \frac{\exp(s_i/T_{\text{sel}})}{\sum_{j \in \text{top-}2k} \exp(s_j/T_{\text{sel}})} \tag{45}$$

$$\text{Stochastic selection:} \quad \text{positions} \sim \text{Multinomial}(\mathbf{p}, k) \tag{46}$$

where $T_{\text{sel}}$ controls the randomness level: lower values favor high-confidence positions (approaching deterministic top-$k$ as $T_{\text{sel}} \to 0$), while higher values increase selection diversity.

`Open-dCoder` experiments use a generation temperature of 0.8, which introduces stochasticity in the token prediction process and the generation process already satisfies the diversity requirements for path search algorithms.

Table 3: Experimental Configurations for Different Tasks of `LLaDA` models.

| Parameter | GSM8K | MATH500 | GPQA | Countdown | Sudoku |
|---|---|---|---|---|---|
| **Task** | | | | | |
| Few-shot examples | 4 | 4 | 5 | 3 | 5 |
| **Generation** | | | | | |
| Generation length | 256 | 1024 | 256 | 128 | 128 |
| Steps | 256 | 1024 | 256 | 128 | 128 |
| Temperature | 0 | 0 | 0 | 0 | 0 |
| CFG scale | 0 | 0 | 0 | 0 | 0 |
| **PC-Sampler** | | | | | |
| $\lambda$ | 0.25 | 0.25 | 0.25 | 0.5 | 0.0 |
| $\alpha$ | 10 | 10 | 10 | 10 | 10 |
| **E-BoN & E-SMC** | | | | | |
| Number of particles | 5 | 5 | 5 | 5 | 5 |
| $\lambda_{\text{weight}}$ | 5.0 | 5.0 | 5.0 | 10.0 | 5.0 |
| Selection temperature | 0.1 | 0.1 | 0.1 | 0.1 | 0.1 |
| **E-SMC** | | | | | |
| Resample interval | 64 | 256 | 64 | 32 | 32 |

Table 4: Experimental Configurations for Different Tasks of `Open-dCoder` model.

| Parameter | HumanEval | HumanEval+ | MBPP | MBPP+ |
|---|---|---|---|---|
| **Generation** | | | | |
| Generation length | 128 | 128 | 128 | 128 |
| Steps | 128 | 128 | 128 | 128 |
| Temperature | 0.8 | 0.8 | 0.8 | 0.8 |
| **E-BoN & E-SMC** | | | | |
| Number of particles | 5 | 5 | 5 | 5 |
| $\lambda_{\text{weight}}$ | 5.0 | 5.0 | 5.0 | 5.0 |
| **E-SMC** | | | | |
| Resample interval | 32 | 32 | 32 | 32 |

### E.3 RESULTS

In this section, we present supplementary experimental results that provide further details supporting the findings discussed in the main text.

Table 5 reports the complete raw data from Figure 3.

Table 6 compares decoding algorithms with Greedy Search, where $c$ represents the number of candidates and $s$ represents the beam size.

Table 7 presents the runtime analysis of different decoding algorithms.

Table 8 provides the detailed numerical values underlying Figure 4.

Figure 6 visually illustrates the evolution of state entropy across different particles throughout the decoding process.

Table 5: Mean $H_{\text{DE}}$, Mean Log PPL, and their Correlation at different steps.

| Step | Mean $H_{\text{DE}}$ | Std $H_{\text{DE}}$ | Mean Log PPL | Correlation |
|------|------|------|------|------|
| 16   | 6.8080 | 0.4320 | 5.731 | 0.891 |
| 32   | 6.1183 | 0.3927 | 5.083 | 0.863 |
| 64   | 5.7352 | 0.3887 | 4.661 | 0.852 |
| 128  | 5.5242 | 0.3823 | 4.396 | 0.853 |
| 256  | 5.3815 | 0.4018 | 4.176 | 0.851 |
| 512  | 5.2318 | 0.3891 | 3.951 | 0.847 |
| 1024 | 5.1197 | 0.3924 | 3.729 | 0.854 |

Table 6: **Comparison with Greedy Search Baselines.** Perplexity is reported for `GPT2-Large` and `Llama-3-8B`. While Greedy Search achieves lower perplexity by restricting the search space, it significantly degrades diversity compared to E-SMC.

| Experiment | Perplexity ↓ | Diversity ↑ |
|------|------|------|
| Vanilla | 68.5 / 66.9 | 5.45 |
| E-BoN ($K = 4$) | 52.3 / 55.2 | 5.39 |
| E-SMC ($K = 4, \Delta i_r = 128$) | 53.4 / 54.8 | **5.46** |
| Greedy Search ($c = 2, s = 1$) | 32.7 / 35.9 | 5.17 |
| Greedy Search ($c = 4, s = 1$) | 19.9 / 24.2 | 4.83 |
| Greedy Search ($c = 8, s = 1$) | 14.0 / 18.7 | 4.59 |
| Greedy Search ($c = 16, s = 1$) | 10.6 / 15.7 | 4.40 |
| Greedy Search ($c = 8, s = 2$) | 11.0 / 15.7 | 4.42 |
| Greedy Search ($c = 8, s = 4$) | 8.8 / 13.7 | 4.09 |
| Greedy Search ($c = 8, s = 8$) | **7.9** / **12.1** | 3.84 |

Table 7: **Runtime analysis comparison.** We report the computation time (in seconds) across different hyperparameter configurations ($S, K, \Delta i_r$) on a single NVIDIA L40 GPU with MDLM. Parallel implementations (**Par.**) reduce latency compared to sequential baselines (**Seq.**) as the number of particles $K$ increases.

| Configuration | | | Baseline | E-BoN | | E-SMC | |
|------|------|------|------|------|------|------|------|
| $S$ | $K$ | $\Delta i_r$ | **Vanilla** | **Seq.** | **Par.** | **Seq.** | **Par.** |
| 256 | 2 | 64 | 7.995 | 13.273 | 8.091 | 13.635 | 8.650 |
| 256 | 4 | 64 | 7.995 | 26.981 | 11.236 | 27.330 | 12.066 |
| 256 | 8 | 64 | 7.995 | 53.718 | 17.943 | 54.617 | 19.329 |
| 1024 | 2 | 256 | 20.256 | 53.848 | 32.790 | 54.652 | 34.644 |
| 1024 | 4 | 256 | 20.256 | 106.605 | 46.928 | 108.457 | 47.771 |
| 1024 | 8 | 256 | 20.256 | 213.582 | 75.134 | 215.317 | 76.474 |

Table 8: Comparison of Sampling Strategies with E-BoN and E-SMC on Code Generation Benchmarks with `Open-dCoder`.

| Strategies | HumanEval | HumanEval+ | MBPP | MBPP+ | Avg.↑ |
|---|---|---|---|---|---|
| P2 | 19.3 | 17.0 | 16.3 | 23.7 | 19.1 |
| *w/* E-BoN | 0.4 ↑ | 0.9 ↑ | 0.5 ↑ | 0.2 ↑ | 0.5 ↑ |
| *w/* E-SMC | 0.2 ↑ | 0.7 ↑ | 0.4 ↑ | 0.3 ↑ | 0.4 ↑ |
| Uniform | 2.7 | 2.6 | 1.1 | 1.6 | 2.0 |
| *w/* E-BoN | 2.2 ↑ | 1.6 ↑ | 1.3 ↑ | 1.5 ↑ | 1.7 ↑ |
| *w/* E-SMC | 1.8 ↑ | 1.3 ↑ | 1.5 ↑ | 1.3 ↑ | 1.5 ↑ |
| Confidence | 7.0 | 6.3 | 2.9 | 5.9 | 5.5 |
| *w/* E-BoN | 3.4 ↑ | 2.7 ↑ | 2.2 ↑ | 3.8 ↑ | 3.1 ↑ |
| *w/* E-SMC | 3.1 ↑ | 2.4 ↑ | 2.1 ↑ | 4.0 ↑ | 2.9 ↑ |
| Entropy | 7.6 | 6.4 | 6.1 | 7.5 | 6.9 |
| *w/* E-BoN | 3.4 ↑ | 3.8 ↑ | 2.7 ↑ | 2.4 ↑ | 3.1 ↑ |
| *w/* E-SMC | 3.7 ↑ | 4.2 ↑ | 3.0 ↑ | 2.6 ↑ | 3.4 ↑ |
| Margin | 4.3 | 3.5 | 2.2 | 3.9 | 3.5 |
| *w/* E-BoN | 3.7 ↑ | 3.3 ↑ | 2.1 ↑ | 2.3 ↑ | 2.8 ↑ |
| *w/* E-SMC | 3.3 ↑ | 3.2 ↑ | 2.2 ↑ | 2.0 ↑ | 2.7 ↑ |

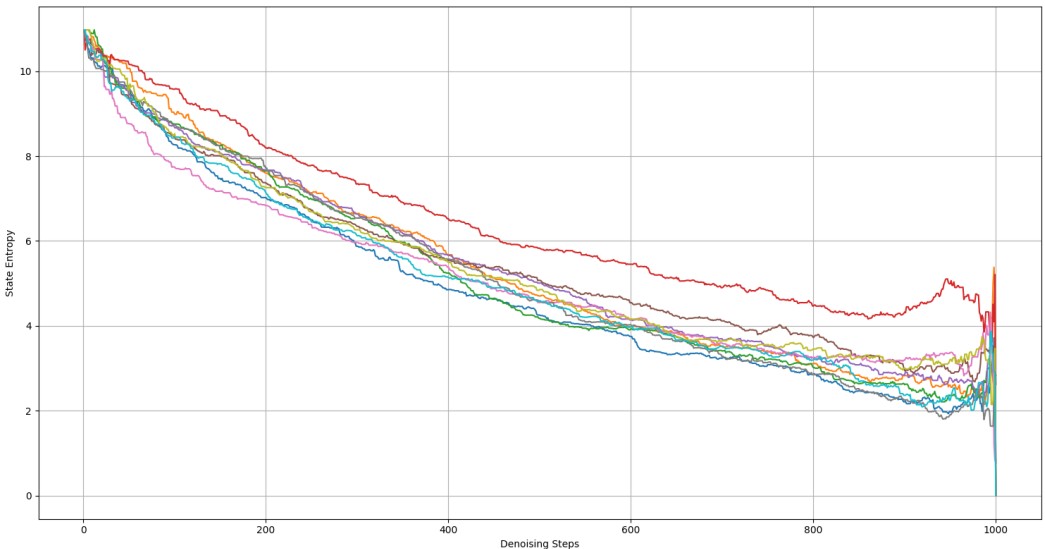

Figure 6: Paths of state entropy over denoising steps for 8 particles, where different colors represent distinct particles. It visually illustrates how the state entropy evolves across each particle throughout the decoding process.

# F  USE OF LARGE LANGUAGE MODELS

Large language models were used solely as a tool to assist in the writing and polishing of this manuscript. They were occasionally employed for tasks such as: i) *refining sentence structure for better readability*; ii) *correcting grammatical errors and typos*; and iii) *polishing the phrasing of certain paragraphs*. The core intellectual content, including research ideas, analyses, experimental designs, and results, did not involve the use of LLMs.

