# OpenReview forum: "Optimizing Decoding Paths in Masked Diffusion Models by Quantifying Uncertainty"
_ICLR.cc/2026/Conference — Submitted to ICLR 2026_

### Official Review · Reviewer_Xwb4 · 2025-10-31

**Soundness:** 3
**Presentation:** 3
**Contribution:** 2
**Rating:** 2
**Confidence:** 5

**Summary:**

The paper defines path uncertainty for MDMs and proposes Denoising Entropy, state entropy, and path entropy. Using these notions of uncertainty, it introduces two path-search methods (greedy selection method / SMC-based method). The authors also provide experimental evidence for these methods.

**Strengths:**

The authors propose principal metrics and simple algorithms to use them for determining the unmasking order of MDMs. These methods indeed surpass the previous heuristic-based unmasking order methods.

**Weaknesses:**

The paper does not present budget-fair comparisons under equal NFE/compute (e.g., matching total denoising function evaluations or wall-clock), making it hard to disentangle algorithmic gains from increased sampling/particle budgets. Although the authors include semi-AR sampling in their baseline, I believe it would've been fairer if the semi-AR + other confidence methods, e.g., semi-AR + prob margin, were also included. Also, the paper's notion of entropy isn't that different from previous papers', in the sense that they still leverage the information of per-position logits. Given all these, I don't think the paper's contribution (at least under the current manuscript) is substantial.

**Questions:**

Please refer to the weakness section!

---

> ### Author Response · Authors · 2025-11-26
> **Rebuttal by Authors**
>
> We appreciate your time and comments. Below, we address the concerns regarding budget fairness, baseline selection, and the novelty of our entropy formulation.
>
> ## 1. Budget-Fair Comparisons [W1]
> > "The paper does not present budget-fair comparisons under equal NFE/compute... making it hard to disentangle algorithmic gains from increased sampling/particle budgets."
>
> We appreciate you pointing out the inadequacy in the computational cost analysis, and we have supplemented the inference timing results in the revised version. However, we respectfully disagree with the premise that the gains are solely due to increased budgets, and we wish to clarify why standard budget matching does not apply straightforwardly to the Pass@1 accuracy metric we report.
>
> - **The Selection Problem in Pass@1:** The core challenge in MDM decoding is not just generating samples, but *selecting* the correct one. Standard baselines lack a global internal metric to evaluate generation quality. Even if we grant a baseline algorithm an increased budget to generate $K$ particles (matching our NFE), the baseline has no mechanism to determine *which* of those $K$ particles to select as the final output for Pass@1 evaluation. In contrast, our contribution is the introduction of **Denoising Entropy** as that internal signal. It allows the model to utilize the increased budget by identifying and selecting the optimal path from the candidates.
>
> - **Comparison with Majority Voting:** We refer to the Budget-Efficiency experiment results in Section 4.4 for explanation. In this part of the discussion, we compare Major Vote with E-BoN/E-SMC. Majority Voting is a standard method for leveraging increased sampling budgets to improve accuracy without an internal quality metric.
>     * **Result:** Under the same computational budget (5 particles), Majority Vote yielded minimal or negative gains, whereas our entropy-guided methods performed better.
>
> - **Runtime Analysis:** We have added a runtime analysis in **Table 7 (Appendix E.3)** of the revision. The main overhead of our algorithm comes from multiple samplings. This overhead can be reduced through parallel implementation.
>
> | Method | Sudoku | Countdown |
> | :---: | :---: | :---: |
> | Majority Vote | -0.8% | +1.0% |
> | E-BoN | +0.6% | +5.9% |
> | E-SMC | +1.6% | +4.1% |
>
> ## 2. Semi-AR + other methods [W2]
> > "I believe it would've been fairer if the semi-AR + other confidence methods... were also included."
>
> We appreciate and acknowledge your suggestion. However, our experimental design was chosen to demonstrate the **universality** and **plug-in** nature of our method, rather than to perform combinatorial search of existing heuristics.
>
> - **Universality Across Strategies:** Our core contribution is demonstrating that Denoising Entropy serves as a robust proxy for Path Uncertainty. To prove this, we provide detailed theoretical proofs and analysis, and apply our method as an enhancer on top of diverse decoding strategies. The fact that our method consistently improves *all* these baselines validates the fundamental effectiveness of the metric, regardless of the underlying decoding order (Semi-AR or otherwise).
>
> - **Adherence to Standard Settings:** We prioritized evaluating baselines in their standard configurations and try to be aligned with previous research.
>     * **Block-wise strategies (Semi-AR, Fast-dLLM):** We utilized the original block size ($L=8$).
>     * **General strategies (Confidence, Margin, EB-Sampler):** We maintained the standard full-sequence setting, as these were not originally designed for block-wise decoding. For instance, the original EB-Sampler conduct experiments in full-sequence setting and notes that semi-autoregressive settings have minimal effect on reasoning tasks like GSM8K (Ben-Hamu et al., 2025).
>
> - **Scope of Contribution:** Combining Semi-AR with other general strategies represents an optimization of existing heuristics. Since determining which generation configuration, combined with specific strategies, performs best on a given task is not our primary concern, we chose to conduct generation and comparisons under a fair setting (i.e., except for strategies specifically designed for block-wise decoding, all strategies are evaluated in full-sequence setting). Our main objective is to demonstrate, through empirical effectiveness across different models, tasks, and decoding methods, that optimizing this specific uncertainty proxy drives quality improvements, which is the core claim of our paper.

---

> ### Author Response · Authors · 2025-11-26
> **Rebuttal by Authors**
>
> ## 3. Novelty of the Entropy Notion [W3]
> > "The paper's notion of entropy isn't that different from previous papers', in the sense that they still leverage the information of per-position logits."
>
> We wish to clarify the distinction between our Denoising Entropy and previous entropy-based metrics.
>
> - **Path vs. Token Uncertainty:** Previous works utilize entropy to measure **local, token-level** uncertainty (e.g., choosing the "easiest" token to unmask next). In contrast, Denoising Entropy is the first metric to formalize and quantify **Path Uncertainty**, which is the cumulative uncertainty of the *entire* non-autoregressive, out-of-order **generation process**. This allows us to model the global path of the generation, which is fundamentally different from token-level uncertainty.
>
> - **Process Modeling for MDMs:** MDMs face a unique challenge compared to autoregressive models: the "decoding path" is not fixed but variable. Our metric is designed for MDMs because it is specifically evaluate the quality of these dynamic, non-sequential paths.
>
> - **Use of Logits is Prerequisite:** We want to note that it is unfair to judge the novelty of an entropy metric based on whether it uses per-position logits, as logits are the building blocks of *any* entropy-based uncertainty quantification. By analogy, Semantic Entropy (Kuhn et al., 2023) also relies on per-position logits, yet it is recognized as a novel uncertainty estimation method because it aggregates those logits to measure semantic consistency rather than lexical confidence. Similarly, our contribution and novelty lies in *how* we aggregate these logits to construct a theoretically grounded proxy for the divergence between the generated path and the true data manifold, a capability that raw per-position logits do not possess.
>
> ---
>
> ## References
>
> [1] Accelerated Sampling from Masked Diffusion Models via Entropy Bounded Unmasking, Ben-Hamu et al., 2025
>
> [2] Semantic Uncertainty: Linguistic Invariances for Uncertainty Estimation in Natural Language Generation, Kuhn et al., 2023
>
> ---
>
> **We hope this addresses your concerns. Please let us know if you have any further questions or you would like additional clarification on any of these points.**

---

> ### Author Response · Authors · 2025-12-03
> **Supplementary Response to W2: Experimental Verification on Semi-AR Combinations**
>
> To address your concern regarding the combination of **Semi-AR** with other decoding methods, we conducted additional experiments on the GSM8K using both LLaDA-Instruct-8B and LLaDA-1.5-8B. We compared the standard Semi-AR baseline (**Confidence**) against Semi-AR combined with **Entropy** and **Margin** constraints across different number of blocks ($1, 2, 8, 32$).
>
> | Model                 | Method         | 1 block  | 2 block | 8 block  | 32 block |
> | :-------------------- | :------------- | :------: | :-----: | :------: | :------: |
> | **LLaDA-Instruct-8B** | **Confidence** |   6.8    |  54.1   |   77.9   | **79.7** |
> |                       | **Entropy**    | **2.2**  |   0.7   |   0.4    |   0.3    |
> |                       | **Margin**     | **11.1** |   5.7   |   5.2    |   4.9    |
> | **LLaDA-1.5-8B**      | **Confidence** |   19.2   |  67.3   | **80.7** |   80.6   |
> |                       | **Entropy**    | **12.1** |   0.8   |   0.8    |   0.8    |
> |                       | **Margin**     | **27.9** |  14.1   |   13.7   |   12.7   |
>
> **Analysis & Observation:**
> The results indicate that simple combinatorial strategies do not yield trivial improvements:
> 1.  **Original Semi-AR Benefits from Block Scaling:** The standard Semi-AR strategy (Confidence) effectively leverages larger number of blocks, achieving significant performance gains (e.g., LLaDA-Instruct-8B improves from 6.8% to 79.7%).
> 2.  **Incompatibility of Heuristics:** Contrary to the expectation that combining methods yields better results, adding token-level confidence constraints (Entropy and Margin) to the block-wise diffusion process introduces negative side effects in our experiments. As the block size increases, these heuristics performance worse (e.g., LLaDA-1.5-8B with Entropy drops to <1% accuracy at 32 blocks).
>
> **Conclusion:**
> This empirical evidence complements our original experimental design. It demonstrates that standard confidence heuristics (Entropy/Margin) may not be natively compatible with block-wise diffusion (Semi-AR) on specific tasks. Consequently, the results presented in Table 2 already reflect the near-optimal performance of these baselines.

---

### Official Review · Reviewer_TKiW · 2025-11-01

**Soundness:** 3
**Presentation:** 3
**Contribution:** 3
**Rating:** 6
**Confidence:** 5

**Summary:**

This paper introduces Denoising Entropy (DE)—a principled metric for quantifying uncertainty in Masked Diffusion Models (MDMs). The authors define State Entropy (hDE) and Path Entropy (HDE) to measure local and cumulative uncertainty along decoding trajectories and propose two corresponding algorithms: Entropy-based Best-of-N (E-BON) and Entropy-guided Sequential Monte Carlo (E-SMC). These methods leverage DE to guide decoding towards more stable, low-uncertainty generative paths. Theoretical analysis establishes hDE as an upper bound on the joint entropy of masked tokens and as a proxy for instantaneous training loss. Experiments show consistent gains in text, reasoning, planning, and code benchmarks.

**Strengths:**

1. **Elegant and Principled Framework.** The formulation of *path uncertainty* and *denoising entropy* is mathematically clean and conceptually satisfying. It connects diffusion decoding with information-theoretic measures in a natural way.
2. **Strong Theoretical Justification.** The proofs that hDE bounds joint entropy and approximates per-token loss are technically sound, providing a rare formal underpinning for an uncertainty metric in MDMs.
3. **Simple but Effective Algorithms.** E-BON and E-SMC are straightforward extensions of existing sampling schemes but grounded in a clear optimization objective. Their connection to sequential Monte Carlo provides interpretability and generality.
4. **Clear Empirical Improvements.** Across multiple reasoning and code benchmarks (GSM8K, GPQA, Sudoku, Countdown, HumanEval), the proposed entropy-guided search yields consistent performance gains over strong baselines such as PC-Sampler and P² sampling.
5. **Broader Implications.** The notion of *path-level uncertainty* could serve as a foundation for calibrated or self-evaluating diffusion decoders, potentially extending to planning, editing, or reinforcement-guided diffusion.

**Weaknesses:**

**Empirical Scope and Novelty of Gains.** While the method consistently improves over baselines, the absolute improvements (typically 1–2% accuracy gains on large reasoning models) may be modest given the added complexity.

**Limited Computational Analysis.** The results primarily focus on accuracy or perplexity; runtime or budget trade-offs for E-SMC versus simpler strategies are not quantified, though SMC is known to be resource-intensive.

**Questions:**

- Does evaluating “path uncertainty” (their Denoising Entropy) require running full decoding simulations, which could be expensive?

- P2 is only shown in Figure 4 but is missed in Figure 2. Can the authors add P2 to Table 2 benchmark?

---

> ### Author Response · Authors · 2025-11-25
> **Rebuttal by Authors**
>
> Thanks for your comments. We appreciate your highlighting the theoretical elegance and mathematical soundness of our path uncertainty framework, as well as the practical effectiveness of our entropy-guided algorithms across diverse reasoning benchmarks.
>
> We address each of your concerns below.
>
> ### 1. Empirical Scope and Novelty of Gains [W1]
>
> > "While the method consistently improves over baselines, the absolute improvements (typically 1–2% accuracy gains on large reasoning models) may be modest given the added complexity."
>
> We appreciate your insightful comment. While we acknowledge that the absolute accuracy gains on reasoning benchmarks are in the range of 1–2%, we believe this magnitude should be interpreted in the context of our experimental design and the fundamental challenge of controlling MDMs. We argue that the consistency of these gains, achieved under highly conservative settings, validates Denoising Entropy as a robust internal signal. And we hope to demonstrate the potential and universality of using Denoising Entropy as an internal signal to improve generation quality through these experiments.
>
> We attribute the nature of the gains to three specific factors:
>
> - **Theoretical Alignment vs. Discrete Accuracy:** Our method is theoretically grounded in minimizing Entropy Drift to align the generated path distribution $\widehat{\Pr}$ with the reference distribution $\Pr$.
> While this alignment significantly improves the likelihood and coherence of the generation (evidenced by the results in Table 1), accuracy on reasoning tasks is a discrete metric.
> Improving the distributional quality increases the probability of correctness, but it does not guarantee a linear conversion to accuracy gains, particularly if the base model lacks the specific knowledge required for the task.
>
> - **Universality Across Strategies:** E-BoN and E-SMC are model-agnostic, training-free enhancers. The gain is consistent across different MDMs, decoding strategies, and tasks. This universality suggests that Denoising Entropy is a robust internal signal in current MDMs.
>
> - **Conservative Local Search Design:** Our objective centered on verifying the efficacy of Denoising Entropy as an internal control signal, rather than prioritizing performance maximization via exploratory mechanisms. As elaborated in Appendix E.2, the baseline LLaDA experiments employ a generation temperature of 0 and to facilitate path search for E-BoN and E-SMC, we incorporated a selection temperature $T_{\text{sel}}$ to introduce controlled stochasticity into the position unmasking phase. In our experiments, $T_{\text{sel}}$ was set to 0.1 to ensure candidate paths stay tightly aligned with the baseline strategy.
>
>
> ### 2. Computational Cost Analysis [W2]
>
> > "The results primarily focus on accuracy or perplexity; runtime or budget trade-offs for E-SMC versus simpler strategies are not quantified, though SMC is known to be resource-intensive."
>
> Thanks for pointing out the computational cost analysis. We have added the results to the revised paper. Here we provide inference timing for MDLM with E-BoN and E-SMC (generated with L=1024). We report the results on a single L40 GPU and a parallel implementation.
>
> | **S**| **K** | **Δi_r** | **Vanilla** | **BoN Sequential** | **BoN Parallel** | **FK Sequential** | **FK Parallel** |
> | ---- | ----- | ------------------ | ----------- | ------------------ | ---------------- | ----------------- | --------------- |
> | 256  | 2     | 64                 | 7.995       | 13.273             | 8.091            | 13.635            | 8.650           |
> | 256  | 4     | 64                 | 7.995       | 26.981             | 11.236           | 27.330            | 12.066          |
> | 256  | 8     | 64                 | 7.995       | 53.718             | 17.943           | 54.617            | 19.329          |
> | 1024 | 2     | 256                | 20.256      | 53.848             | 32.790           | 54.652            | 34.644          |
> | 1024 | 4     | 256                | 20.256      | 106.605            | 46.928           | 108.457           | 47.771          |
> | 1024 | 8     | 256                | 20.256      | 213.582            | 75.134           | 215.317           | 76.474          |
>
> **Analysis of the results:** As shown in the table above, the inference timing of E-BoN and E-SMC are relatively close and the primary overhead of the algorithm comes from multiple samplings. The parallel implementation of the algorithm can significantly reduce the inference time.
>
> **Add Diversity Metric to Evaluate the Quality of Generation:** We also added the diversity metric to the revised paper in Table 1 to further demonstrate the improvement of generation quality brought by our algorithm.

---

> ### Author Response · Authors · 2025-11-25
> **Rebuttal by Authors**
>
> ### 3. Simulations about Evaluating Path Entropy [Q1]
>
> > "Does evaluating “path uncertainty” (their Denoising Entropy) require running full decoding simulations, which could be expensive?"
>
> We appreciate this question regarding the computational expense of our method.
>
> **Short Answer:** While calculating the final Path Entropy ($H_\text{DE}$) for a sequence does require traversing a decoding path, the metric itself is computed **online** as an internal signal during generation. It does not require separate, external model runs or post-hoc simulations.
>
> We offer the following detailed breakdown:
>
> **The Metric Calculation Introduces Minimal Overhead:** Calculating State Entropy ($h_\text{DE}$) adds almost no computational cost, negligible compared to model inference.
> - **Internal Signal:** The metric is derived directly from the model's predictive probability distributions, $\hat{p}_{\theta}(X_0|z_t)$, which the MDM **already computes** at every step to perform denoising.
> - **No Extra Forward Passes:** Calculating the entropy of these existing distributions is a simple mathematical operation that does not require additional forward passes or external reference models.
>
> **Cost Depends on the Number of Search Candidates:** For both proposed algorithms, the primary computational expense arises from the multi-sampling strategy used to explore the path space, scaling linearly with the number of particles/paths
> - **E-BON:** We acknowledge that E-BON requires generating $K$ full candidate paths to select the best one. This indeed scales the cost linearly with the number of particles.
> - **E-SMC:** While this method also operates on $K$ parallel particles, it results in only a minimal increase in sampling time compared to E-BON. The additional steps introduced by E-SMC mainly involve the evaluation of potentials and the resampling of indices, which are computationally inexpensive operations compared to the MDM inference.
>
> As shown in the experimental results we presented above, the main overhead of the algorithm still comes from multiple samplings.
>
>
> ### 4. Add P2 results to Table 2 [Q2]
>
> > "P2 is only shown in Figure 4 but is missed in Figure 2. Can the authors add P2 to Table 2 benchmark?"
>
> | Model | GSM8K | MATH500 | GPQA | Countdown | Sudoku | Avg. $\uparrow$ |
> | :--- | :--- | :--- | :--- | :--- | :--- | :--- |
> | **LLaDA-Instruct-8B** | 11.6 | 3.4 | 28.2 | 34.2 | 24.0 | 20.3 |
> | **LLaDA-1.5-8B** | 25.4 | 5.6 | 28.8 | 33.6 | 27.8 | 24.2 |
>
> **Action:** Thanks for your suggestion. P2 represents a significant re-masking decoding strategy. In each iteration, it sorts tokens by their global confidence scores, dynamically discarding the least reliable tokens (by resetting them to masks) while filling the most probable slots. We have incorporated the performance of P2 into **Table 2** of the revised paper.
>
> ---
>
> **Thanks again for your insightful comment and feedback. Please let us know if there is any additional information we can include that would be helpful for your review.**

---

> > ### Comment · Reviewer_TKiW · 2025-11-26
> >
> > thank you for the response. I'll keep my score.

---

### Official Review · Reviewer_nHrU · 2025-11-04

**Soundness:** 2
**Presentation:** 2
**Contribution:** 2
**Rating:** 4
**Confidence:** 4

**Summary:**

Masked Diffusion Models (MDMs) offer multiple sampling paths during inference, unlike Autoregressive Models. Prior work has shown that choosing the correct sampling path can significantly affect performance, but most existing methods rely on locally greedy strategies. This paper proposes optimizing the decoding path using denoising entropy, defined as the average of the entropies of all masked tokens along the path. To achieve this, the method maintains K particles (candidate paths) during generation and steers them toward low-entropy paths using Sequential Monte Carlo and then ultimately selects the best candidate path according to the denoising entropy. The paper provides a theoretical explanation of the approach and presents extensive experiments on language modeling and several reasoning benchmarks.

**Strengths:**

- The paper studies an important problem — optimizing the decoding path in Masked Diffusion Models — and introduces a relatively simple method that scales inference-time computation to achieve notable performance gains.

- The paper also provides extensive experiments evaluating the proposed method across a variety of tasks. It is also nice that the paper attempts to offer a theoretical justification for the proposed approach (though, as noted below, there are some issues with it).

**Weaknesses:**

- Propositions 1 and 2 rely on several assumptions that require further justification. In particular, both propositions appear to assume $p_{\theta}(x_0^{\ell} | z_t) = q(x_0^{\ell} | z_t)$. This is a strong assumption in the context of optimizing decoding paths for MDMs. If the learned posterior equals the true data posterior, then all decoding paths would yield the same distribution (see page 7 in [1] for an explanation). This makes the assumption a bit unrealistic, and the practical implications of the results remain unclear.

- The paper’s writing could be improved in several places. For instance, the proof of Proposition 1 (lines 825–830) seems to rely on the assumption that the learned posterior equals the data posterior, but this assumption is not explicitly stated in Proposition 1. Additionally, the oracle state entropy is defined in the appendix but is used in Proposition 1. There are also minor typos (e.g., “L” on line 803).

- As pointed out in [2], MDMs can exploit the generative perplexity metric by repeating high-frequency words. Therefore, the entropy metric from [2] should also be reported in Table 1 to ensure that the proposed method is not exploiting this issue.

- For decoding strategies such as confidence, margin, and EB-sampler, it is standard practice to use a smaller block size (e.g., 64) and apply these strategies within each block — as done in the LLaDA paper. However, this paper uses the full block size, which likely causes a significant performance drop for the baselines.

- Most of the performance gains in the reasoning and planning experiments come from the Countdown task, while other tasks show only marginal improvements.

[1] Train for the Worst, Plan for the Best: Understanding Token Ordering in Masked Diffusions

[2] Masked Diffusion Models are Secretly Time-Agnostic Masked Models and Exploit Inaccurate Categorical Sampling

**Questions:**

See weaknesses section.

---

> ### Author Response · Authors · 2025-11-25
> **Rebuttal by Authors**
>
> Thank you for noting our method's simplicity, experimental gains and theoretical soundness. We value the observation regarding the distinction between the true posterior and the learned posterior, as well as the suggestion to evaluate diversity.
>
> Below, we address your concerns point by point.
>
> ### 1. Theoretical Assumptions and Clarifications [W1, W2]
>
> > "Propositions 1 and 2 rely on several assumptions that require further justification. In particular, both propositions appear to assume $p_\theta(x_\theta^l|z_t)=q(x_\theta^l|z_t)$. This is a strong assumption in the context of optimizing decoding paths for MDMs..."
>
> > "the proof of Proposition 1 (lines 825–830) seems to rely on the assumption that the learned posterior equals the data posterior, but this assumption is not explicitly stated in Proposition 1. Additionally, the oracle state entropy is defined in the appendix but is used in Proposition 1. There are also minor typos (e.g., “L” on line 803)."
>
> We appreciate that you pointed out the ambiguity regarding the assumptions in our propositions. We clarify the distinct assumptions for each:
>
> **Proposition 1:** We clarify that this proposition does not rely on the assumption that the learned posterior equals the true data posterior ($p_\theta(x|z_t) = q(x|z_t)$). The confusion may come from our imprecise phrasing in the initial submission, where we referred to the _true joint posterior distribution_. We have corrected the definition to explicitly state that Oracle State Uncertainty refers to the joint predictive distribution of the model itself. Under this definition, $h_{\text{DE}}$ is a valid upper bound for the model's own joint uncertainty, independent of the true data distribution.
>
> **Proposition 2:** We acknowledge that assuming strict equality ($p_\theta \approx q$) is too strong.
> - **Refinement:** And we have refined the theoretical framing by introducing the concept of an $\epsilon$-accurate model. We show that as long as the divergence between the model and the true posterior is bounded by $\epsilon$ (where $\epsilon \to 0$ as training converges), $h_{\text{DE}}$ serves as a valid proxy for the normalized training loss.
> $$
> \underbrace{\frac{1}{|\mathcal{M} _ t|} \sum _ {\ell \in \mathcal{M} _ t} \left( -\log p_{\theta}(x _ 0^\ell | \mathbf{z} _ t, t) \right)} _ {\text{training loss}}-h _ {\texttt{DE}}(\mathbf{z} _ t) \leq \epsilon+\sqrt{\frac{\epsilon}{2}}logK
> $$
> - **New Analysis (Entropy Gap):** To further address the implication of these results, we have added a new derivation linking entropy to generation quality. We derive a lower bound for the KL divergence between the true path distribution $\Pr$ and the generated distribution $\widehat{\Pr}$:
> $$
> D_{\mathrm{KL}}(\Pr \| \widehat{\Pr}) \geq \frac{1}{2B^2} (\mu_{\widehat{\Pr}} - \mu_{\Pr})^2
> $$
> This inequality demonstrates that minimizing the Entropy Gap $|\mu_{\widehat{\Pr}} - \mu_{\Pr}|$ is a necessary condition for aligning the generated distribution with the true distribution. And we show that our algorithms (E-BoN/E-SMC) improve quality by explicitly tightening this bound.
>
> **Action 1:** For Oracle State Uncertainty definition, we have moved this formal definition to **Section 3.2** in the main text and have corrected the typos. We also show the lower bound for the KL divergence between the true path distribution $\Pr$ and the generated distribution $\widehat{\Pr}$ in **Proposition 3** and detailed derivations in **Appendix C**.
>
> ### 2. Diversity and Exploitation of Perplexity [W3]
>
> > "...MDMs can exploit the generative perplexity metric by repeating high-frequency words. Therefore, the entropy metric should also be reported to ensure that the proposed method is not exploiting this issue."
>
> **Add diversity metric to evaluate the quality of generation:** Thanks for pointing out that perplexity alone can sometimes be hacked by repetition, and thus using perplexity as the sole metric to represent generation quality may lack sufficient validity. To verify that our method does not exploit this issue, we adopted the metric described in the referenced paper (Zhang et al., 2024), evaluated the Diversity (consistent with the sentence entropy defined in the referenced paper) of the generated outputs, and present the results in **Table 1**. We also compare with greedy search, which demonstrates the repetition issue.
>
> **Analysis of the results:** The results demonstrate that E-BoN and E-SMC can reduce perplexity while preserving diversity, unlike greedy search which can reduce perplexity but at the cost of causing repetition problems and thus reducing the quality of generation. Under specific configurations, E-SMC can improve generation quality without compromising diversity.

---

> ### Author Response · Authors · 2025-11-25
> **Rebuttal by Authors**
>
> | Strategy | PPL (Llama-3-8B) | Diversity |
> | :--- | :---: | :---: |
> | **Vanilla** | 66.9 | 5.45 |
> | **E-BoN (K=4)** | 55.2 | 5.39 |
> | **E-SMC (K=4, Δi_r=128)** | 54.8 | **5.46** |
> | Greedy Search (num_candidates=8, beam_size=1)  |   18.7   |   4.59    |
> | Greedy Search (num_candidates=8, beam_size=2)  |   15.7   |   4.42    |
> | Greedy Search (num_candidates=8, beam_size=4)  |   13.7   |   4.09    |
> | Greedy Search (num_candidates=8, beam_size=8)  | **12.1** |   3.84    |
>
> ### 3. Baseline Implementation Details [W4]
>
> > "For decoding strategies such as confidence, margin, and EB-sampler, it is standard practice to use a smaller block size (e.g., 64) and apply these strategies within each block... However, this paper uses the full block size, which likely causes a significant performance drop for the baselines."
>
> We provide the following clarifications and justifications for our experimental setup:
>
> **Differentiated Settings:** We categorize the baselines into two types: those explicitly designed for semi-autoregressive generation and general non-autoregressive strategies.
> - **Block-wise strategies:** For **Semi-AR** and **Fast-dLLM**, we did utilize a block size of 8 as original settings.
> - **General strategies:** For general strategies (e.g., Confidence, Margin, EB-Sampler), we maintained the standard full-sequence non-autoregressive setting.
>
> **Adherence to Original Methodologies:** General decoding strategies were not originally designed specifically for block-wise decoding. For instance, the original **EB-Sampler** paper primarily evaluates using full generation length. While they explore semi-autoregressive settings as an additional ablation, they note that it has **minimal effect** on performance for reasoning tasks such as GSM8K. Therefore, evaluating these baselines in the full-sequence setting aligns with their original design and standard usage.
>
> **Plug-in Versatility:** The contribution of our entropy-based guidance is their role as **universal plug-in enhancers** rather than standalone decoding strategies.
> - In **Table 2**, we focused on the state-of-the-art **PC-Sampler** to demonstrate our method's efficacy on top of the strongest baseline.
> - In **Figure 4** and **Table 8**, we applied our method across different decoding strategies.
>
> **Objective:** These experiments more focus on the potential and universality of using Denoising Entropy as an internal signal to improve generation quality through these experiments.
>
> **Action 2:** We have updated **Section 4.3** in the revision to explicitly list the block size ($L=8$) for Semi-AR and Fast-dLLM to prevent confusion.

---

> ### Author Response · Authors · 2025-11-25
> **Rebuttal by Authors**
>
> ### 4. Improvements [W4]
>
> > "Most of the performance gains in the reasoning and planning experiments come from the Countdown task, while other tasks show only marginal improvements."
>
> While the gains on the **Countdown** task are indeed the most pronounced (likely due to its high sensitivity to planning), we note that our method shows consistent improvements across different models, sampling strategies and tasks. We believe these consistent gains on challenging math and logic tasks demonstrate that the benefit of path optimization extends well beyond the specific dynamics of Countdown. We attribute the magnitude of these gains to three key factors:
>
> - **Distributional Alignment vs. Accuracy:** Theoretically, our method minimizes the entropy gap to align the generated path distribution $\widehat{\Pr}$ with the true distribution $\Pr$ (as discussed in the response to W1). While this improves generation *quality* and *likelihood*, task accuracy is a discrete metric. A better distribution increases the probability of correctness but does not guarantee a linear conversion to accuracy gains across all task types. Previous research (Agarwal et al., 2025) that improved benchmark accuracy by entropy minimization are also more of an empirical finding rather than a theoretical guarantee.
>
> - **Model Intrinsic Capability:** Improvements in decoding strategies can optimize the utilization of a model's latent knowledge, but they cannot create capabilities absent from the base model. Given the difficulty of the reasoning benchmarks for non-autoregressive models, the base model's capability acts as a performance ceiling. The effectiveness of previous entropy minimization methods is also based on the assumption that "if a model is reasonably capable, it is more likely to be correct when it is confident." Therefore, the specific improvement is closely related to the benefit that the base model can bring through confidence on specific tasks.
>
> - **Conservative Exploration ($T_{\text{sel}} = 0.1$):** Our primary goal was to validate the effectiveness of **Denoising Entropy** as a internal control signal, rather than to maximize performance through aggressive exploration. As detailed in **Appendix D.2**, the base LLaDA experiments use a generation temperature of 0, making token prediction deterministic. To enable path search for E-BoN and E-SMC, we introduced a **selection temperature** ($T_{\text{sel}}$) to inject controlled randomness into the position unmasking step. We set $T_{\text{sel}}$ to a very low value of **0.1**. This creates a "local search" effect where candidate paths remain very close to the baseline strategy. The fact that we achieve consistent improvements despite this minimal variation validates that $H_{\text{DE}}$ correctly identifies high-quality paths within a narrow search space.
>
> ---
>
> ## References
>
> [1] Masked Diffusion Models are Secretly Time-Agnostic Masked Models and Exploit Inaccurate Categorical Sampling, Zheng et al., 2024
>
> [2] Accelerated Sampling from Masked Diffusion Models via Entropy Bounded Unmasking, Ben-Hamu et al., 2025
>
> [3] The Unreasonable Effectiveness of Entropy Minimization in LLM Reasoning, Agarwal et al., 2025
>
> ---
>
> **Thank you again for your valuable feedback, which we have incorporated into the revised manuscript. We believe these changes have substantially improved our work and are grateful for your guidance. We look forward to hearing any further thoughts and comments you may have.**

---

> ### Author Response · Authors · 2025-12-03
> **Supplementary Response to W4: Empirical Verification on Block Size Settings**
>
> To analyze the impact of the block-wise approach, we conducted additional experiments on GSM8K. We compared the standard Semi-AR baseline (**Confidence**) against **Entropy** and **Margin** across varying number of blocks ($L=1, 2, 8, 32$) to observe their behavior under semi-autoregressive settings.
>
> | Model | Method | 1 block | 2 block | 8 block | 32 block |
> | :--- | :--- | :---: | :---: | :---: | :---: |
> | **LLaDA-Instruct-8B** | **Confidence** | 6.8 | 54.1 | 77.9 | **79.7** |
> | | **Entropy** | **2.2** | 0.7 | 0.4 | 0.3 |
> | | **Margin** | **11.1** | 5.7 | 5.2 | 4.9 |
> | **LLaDA-1.5-8B** | **Confidence** | 19.2 | 67.3 | **80.7** | 80.6 |
> | | **Entropy** | **12.1** | 0.8 | 0.8 | 0.8 |
> | | **Margin** | **27.9** | 14.1 | 13.7 | 12.7 |
>
> **Analysis & Observation:**
> We observed from our experimental results contradict the assumption that block-wise application benefits these heuristics:
> 1.  **Divergent Behaviors:** While the standard Semi-AR strategy (**Confidence**) benefits from scaling up the number of blocks (e.g., LLaDA-Instruct-8B improves from 6.8% to 79.7%), the other heuristics show the opposite trend.
> 2.  **Performance Degradation in Semi-AR:** Applying **Entropy** and **Margin** constraints within a block-wise framework introduces negative side effects. As the number of blocks increases, the performance of these methods drops (e.g., Margin drops from 11.1% to 4.9% on LLaDA-Instruct-8B).
>
> **Conclusion:**
> This empirical evidence suggests that the poor performance of these baselines is not due to the lack of block-wise implementation (Semi-AR). In fact, forcing these strategies into the semi-autoregressive setting (as analyzed above) yields worse results compared to the settings presented in Table 2.

---

### Official Review · Reviewer_3HRL · 2025-11-05

**Soundness:** 3
**Presentation:** 3
**Contribution:** 3
**Rating:** 4
**Confidence:** 4

**Summary:**

This paper studies the problem of decoding from masked diffusion models (MDMs). They make the observation that the quality of the final output is highly sensitive to the order with which the tokens are decoded in an MDM. The authors attribute the variability in output quality to the cumulative predictive uncertainty along the different generative paths. In an attempt to quantify such uncertainty, the authors introduce *denoising entropy*. The authors then develop two decoding algorithms designed to optimize the decoding path based on *denoising entropy*, a post-hoc filtering approach as well as a real-time guidance strategy. They find that the proposed entropy-guided decoding significantly improve generation quality.

**Strengths:**

- The paper is fairly well-written with an abundance of figure to aid with the communication of ideas

- The authors show that their approaches E-BoN and E-SMC manage to increase the average accuracy when paired with any suite of decoding approaches.

- The authors provide a theoretical justification for their newly proposed decoding approach

- The experiments cover a range of tasks and models

**Weaknesses:**

- The authors use GPT2 to evaluate the perplexity of generations. I expected to see the perplexity reported using a much more capable LM given the current landscape.

- As the authors might know, in LLMs, greedy decoding and beam search, which by definition attempts to approximate the lowest-uncertainty generation path, tends to perform quite poorly and are typically avoided as a decoding approach. I would've expected a discussion of how one is to consolidate these well established finding in the autoregressive language modeling community with the findings of the paper.

- Given the argument that authors are attempting to make, I would have expected an impractical beamsearch baseline that (potentially) always upperbounds their more tractable entropy criterion.

- Typos:
-> line 029 "Unlike ARMs rely"

**Questions:**

- In the introduction, "High cumulative uncertainty harms output consistency, while low uncertainty indicates reliable paths". Is this a heuristic that has been shown to work in practice and has been established apriori, a heuristic that the authors are putting forth, or something that can be ascertained?

- Could you please walk me through the definition of the *Oracle State Uncertainty* and why it is intractable to compute? I would have expected us to be more interested in deriving bounds on the entropy/uncertainty of a decoding path (I also find the bound perhaps too loose to be very useful, unless the authors manage to convince me otherwise). Furthermore, proposition 2 only makes the case that state entropy is a proxy for the MDM loss, but fails to make a strong case (to my understanding) for why it is a good criterion to optimize for while decoding. I also refer the authors to [1] which studies the delicate balance between accuracy and calibration in LLMs (which might potentially be applicable in MDMs)

-Have the authors considered running a form of beam search to further validate their claims that states with the lowest entropy have the highest quality? It could perhaps be a more expensive upper bound of their approach, and would make for a more convincing argument.

References:

[1] Mark Braverman, Xi Chen, Sham Kakade, Karthik Narasimhan, Chiyuan Zhang, & Yuhuai Zhang. (2020). Calibration, Entropy Rates, and Memory in Language Models. In Proceedings of the 37th International Conference on Machine Learning (ICML 2020).

---

> ### Author Response · Authors · 2025-11-25
> **Rebuttal by Authors**
>
> We appreciate the time you took to review our work and the helpful feedback you provided. We were glad to hear that you recognized the significance of the problem we address and found our theoretical justification for the decoding approach to be sound. We are also grateful for your suggestion to connect our work with the autoregressive model perspective, which has helped us strengthen the theoretical link between denoising entropy and generation quality.
>
> ## 1. Evaluation with Larger Models [W1]
>
> > "The authors use GPT2 to evaluate the perplexity of generations. I expected to see the perplexity reported using a much more capable LM given the current landscape."
>
> **Fair Comparison Baseline:** We initially selected GPT-2-Large as the evaluator because its parameter count is comparable to the MDLM used in our experiments. Furthermore, it serves as the standard evaluator in the original MDLM Paper (Sahoo et al., 2024), ensuring our results are directly comparable to prior work.
>
> **Extended Evaluation with Llama-3:** We agree that evaluating with a much more capable model provides a more robust assessment. Following your suggestion, we conducted additional evaluations using **Llama-3-8B** (approximately 10x the parameters of GPT-2-Large and 40x of the MDLM). As shown in the updated **Table 1**, while the absolute perplexity values differ, the strong positive correlation between Path Entropy and generative perplexity remains consistent. E-SMC continues to consistently show the optimal performance under this evaluation.
>
> | S | K | Δi_r | Vanilla | E-BoN | E-SMC |
> | :---: | :---: | :---: | :---: | :---: | :---: |
> | 128 | 4 | 32 | 84.5 | 66.1 | **61.9** |
> | 256 | 4 | 32 | 66.9 | 55.2 | **51.5** |
> | 128 | 4 | 32 | 84.5 | 66.1 | **61.9** |
> | 128 | 8 | 32 | 84.5 | 59.8 | **59.0** |
> | 256 | 2 | 32 | 66.9 | 59.1 | **57.0** |
> | 256 | 4 | 32 | 66.9 | 55.2 | **51.5** |
> | 256 | 8 | 32 | 66.9 | 51.1 | **45.1** |
> | 256 | 4 | 8 | 66.9 | 55.2 | **44.4** |
> | 256 | 4 | 16 | 66.9 | 55.2 | **48.1** |
> | 256 | 4 | 32 | 66.9 | 55.2 | **51.5** |
> | 256 | 4 | 64 | 66.9 | 55.2 | **51.8** |
> | 256 | 4 | 128 | 66.9 | 55.2 | **54.1** |
>
> **Action 1:** We have updated **Table 1** and the corresponding analysis in the revised draft to include perplexity results computed by Llama-3-8B.
>
> ## 2. Relation between Denoising Entropy and Generation Quality [Q1, Q2.2, Q2.3]
>
> > "High cumulative uncertainty harms output consistency... Is this a heuristic... or something that can be ascertained?"
>
> > "I would have expected us to be more interested in deriving bounds on the entropy/uncertainty of a decoding path"
>
> > "Proposition 2... fails to make a strong case for why it is a good criterion to optimize for while decoding."
>
> **Empirical Evidence:** The statement is grounded in the established observation that capable language models are generally more likely to be correct when they are confident. Entropy minimization, whether as a training objective or an inference-time optimization, has been empirically proven to enhance performance on challenging tasks in recent autoregressive language model research (Agarwal et al., 2025; Gao et al., 2025).
>
> **Theoretical Justification:** We provide a justification for why optimizing entropy improves quality. We define generation quality as the KL divergence $D_{\mathrm{KL}}(\Pr \| \widehat{\Pr})$ between the true path distribution $\Pr(\tau)$ and the model's predicted distribution $\widehat{\Pr}(\tau)$. Let $\mu_{\Pr} = E_{\tau \sim \Pr}[H_{\texttt{DE}}(\tau)]$ be the expected Path Entropy on reference paths, and $\mu_{\widehat{\Pr}} = E_{\tau \sim \widehat{\Pr}}[H_{\texttt{DE}}(\tau)]$ be the expected Path Entropy of the model's generations. We derive the following lower bound (**Proposition 3** in the revision):
> $$
> D_{\mathrm{KL}}(\Pr \| \widehat{\Pr}) \geq \frac{1}{2B^2} \left( \mu_{\widehat{\Pr}} - \mu_{\Pr} \right)^2
> $$
> This inequality demonstrates that minimizing the gap between the generated entropy and the reference entropy is a necessary condition for minimizing the divergence from the true distribution. Since autonomous generations typically exhibit Entropy Drift, where $\mu_{\widehat{\Pr}} > \mu_{\Pr}$, minimizing $\mu_{\widehat{\Pr}}$ via E-BoN or E-SMC tightens this lower bound, thereby theoretically improving generation quality.
>
> **Action 2:** We have expanded **Section 3.4** to explicitly include this analysis, clarifying that E-BoN and E-SMC optimize generation by minimizing the entropy gap $\left| \mu_{\widehat{\Pr}} - \mu_{\Pr} \right|$. Detailed derivations are provided in **Appendix C**.

---

> ### Author Response · Authors · 2025-11-25
> **Rebuttal by Authors**
>
> ## 3. Comparison with Greedy Search and Diversity [W2, W3, Q3]
>
> > "I would have expected an impractical beam search baseline that (potentially) always upperbounds their more tractable entropy criterion."
>
> > "...considered running a form of beam search to further validate their claims that states with the lowest entropy have the highest quality?"
>
> **The Risk of Over-Optimization:** We appreciate the suggestion to discuss greedy decoding and beam search. Both prior studies and our findings suggest that within appropriate bounds, low entropy is associated with higher generation quality. However, this holds only under the premise that $\mu_{\widehat{\Pr}} \geq \mu_{\Pr}$. Once $\mu_{\widehat{\Pr}} < \mu_{\Pr}$, lower entropy instead corresponds to a larger lower bound of $D_{\mathrm{KL}}(\Pr \| \widehat{\Pr})$.
>
> Over-optimization means entropy is pushed significantly below the reference level ($\mu_{\widehat{\Pr}} < \mu_{\Pr}$), which increases the lower bound of the KL divergence (as shown in the equation above) and leads to degradation. We also demonstrate that $\widehat{\Pr} _ {\texttt{E-BoN}}(\tau)$ and $\widehat{\Pr} _ {\texttt{E-SMC}}(\tau)$ do not suffer from over-optimization issues, and for E-SMC, we prove the existence of an optimal $\lambda ^ *$ such that $\mu _ {\widehat{\Pr} _ {\texttt{E-SMC}}}(\lambda ^ *) = \mu _ {\Pr}$, details in **Appendix C.3**. Over-optimization emerges in LLMs as the repetition trap, where greedy decoding and beam search yield extremely low entropy but generate repetitive, low-quality text. Reviewer nHrU also mentioned a research reveals that MDMs can exploit generative perplexity metric by repeating high-frequency words (Zhang et al., 2024). We referred to the diversity metric in the this research to measure the quality of generation.
>
> **Diversity vs. Perplexity Analysis:** We compared our methods against greedy search using both perplexity and diversity. For a sequence of length $L$ that contains $K$ distinct tokens, with each token $k$ occurring $L_k$ times. $p_k = \frac{L_k}{L}$ represents the probability of occurrence of token $k$. The diversity $H$ is computed as $H = -\sum_{k=1}^{K} p_k \log p_k$.
>
> | Strategy | PPL (Llama-3-8B) | Diversity | Interpretation |
> | :--- | :---: | :---: | :--- |
> | **Vanilla** | 66.9 | 5.45 | High diversity, high PPL |
> | **E-BoN (K=4)** | 55.2 | 5.39 | **Balanced** |
> | **E-SMC (K=4, Δi_r=128)** | 54.8 | **5.46** | **Balanced** |
> | Greedy Search (num_candidates=2, beam_size=1)  |   35.9   |   5.17    | |
> | Greedy Search (num_candidates=4, beam_size=1)  |   24.2   |   4.83    | Signs of repetition |
> | Greedy Search (num_candidates=8, beam_size=1)  |   18.7   |   4.59    | |
> | Greedy Search (num_candidates=8, beam_size=2)  |   15.7   |   4.42    | |
> | Greedy Search (num_candidates=8, beam_size=4)  |   13.7   |   4.09    | |
> | Greedy Search (num_candidates=8, beam_size=8)  | **12.1** |   3.84    | **Severe collapse** |
>
> As shown, Gready Search achieves significantly low PPL by sacrificing diversity (collapsing to 3.84), indicating degenerate repetition. In contrast, E-BoN and E-SMC reduce PPL while maintaining diversity comparable to the Vanilla baseline. This confirms our methods optimize entropy effectively without falling into the degenerate modes typical of greedy/beam search.
>
> **Action 3:** We have added diversity results to **Table 1** and included a discussion in the **Appendix E** comparing our approach to Greedy Search to clarify why simply maximizing probability (minimizing entropy indefinitely) is suboptimal.
>
> ## 4. Definition of Oracle State Uncertainty [Q2.1]
>
> > "Could you please walk me through the definition of the Oracle State Uncertainty..."
>
> **Clarification:** We note that the definition of Oracle State Uncertainty was placed in the Appendix but referenced in Proposition 1, which may lead to ambiguity. To clarify, **Oracle State Uncertainty** is defined as the entropy of the model's joint predictive distribution over the masked tokens, $H(p_\theta(X_{\mathcal{M}_t} | z_t, t))$. It represents the ideal uncertainty metric that accounts for the correlations and dependencies between all masked tokens. Specifically, under the current state, this Oracle State Uncertainty is calculated based on the probability distribution of all possible combinations of tokens across all masked positions over the vocabulary. It is intractable because calculating the joint predictive distribution requires summing over all possible completions ($V^{|\mathcal{M}_t|}$).
>
> **Action 4:** We have moved the formal definition of Oracle State Uncertainty from the Appendix to **Section 3.2** before Proposition 1 to ensure the clarity.
>
> ## 5. Typos [W4]
>
> > "Typos: -> line 029 'Unlike ARMs rely'"
>
> **Action 5:** Thanks for pointing out the typo. We have corrected it in the revised draft.

---

> ### Author Response · Authors · 2025-11-25
> **Rebuttal by Authors**
>
> ## References
>
> [1] Simple and Effective Masked Diffusion Language Models, Sahoo et al., 2024
>
> [2] The Unreasonable Effectiveness of Entropy Minimization in LLM Reasoning, Agarwal et al., 2025
>
> [3] One-shot Entropy Minimization, Gao et al., 2025
>
> [4] Masked Diffusion Models are Secretly Time-Agnostic Masked Models and Exploit Inaccurate Categorical Sampling, Zheng et al., 2024
>
> ---
>
> **Thank you for your insightful comments, and for referring us to explore the balance between accuracy and calibration in LLMs. Your feedback has been instrumental in helping us refine our analysis. If you have any additional comments or suggestions, please feel free to let us know.**

---

### Author Response · Authors · 2025-12-03
**Overall Response to All Reviewers**

We sincerely thank all reviewers for their thoughtful feedback and constructive suggestions. We are encouraged that the reviewers recognized the elegance and mathematical soundness of our path uncertainty framework, the consistent empirical improvements across diverse tasks and models, and the simplicity and generality of E-BoN and E-SMC as training-free plug-in enhancers. Below, we summarize the main improvements made in response to the reviewers' comments:

### 1. Enhanced Theoretical Foundation

- **Stronger Justification for Entropy Optimization [Reviewer `3HRL` `nHrU`]:** We introduce **Proposition 3**, establishing a lower bound $D_{\mathrm{KL}}(\Pr \| \widehat{\Pr}) \geq \frac{1}{2B^2} (\mu_{\widehat{\Pr}} - \mu_{\Pr})^2$ that rigorously justifies why minimizing the Entropy Gap improves generation quality (detailed in **Appendix C**). We also refined the assumptions underlying Propositions 1 and 2 by introducing $\epsilon$-accurate models, and moved the Oracle State Uncertainty definition to **Section 3.2** for clarity.

### 2. More Comprehensive Evaluation

- **Evaluation with Larger Models [Reviewer `3HRL`]:** We conducted additional evaluations using **Llama-3-8B** (40× larger than MDLM) and report results in **Table 1**, confirming the strong correlation between Path Entropy and perplexity remains consistent.

- **Diversity Analysis [Reviewer `3HRL` `nHrU` `TKiw`]:** We added diversity measurements showing E-BoN and E-SMC maintain diversity (5.39-5.46) comparable to vanilla (5.45), unlike greedy/beam search which collapses to 3.84 in **Table 1**.

- **Computational Cost Analysis [Reviewer `TKiw` `Xwb4`]:** We added inference timing results showing the overhead primarily comes from multiple samplings and can be reduced via parallel implementation in **Table 7, Appendix E.3**.

- **Semi-AR Combination Experiments [Reviewer `Xwb4`]:** Additional experiments combining Semi-AR with token-level confidence methods (Entropy/Margin) reveal that these combinations introduce negative side effects at larger block sizes, complementing our original baseline configurations.

### 3. Clarified Experimental Design

- **Baseline Configurations [Reviewer `nHrU` `Xwb4`]:** We documented all baseline settings in **Section 4.3**, clarifying that block-wise strategies use $L=8$ blocks while general strategies use standard full-sequence settings.

- **Comparison with Greedy/Beam Search [Reviewer `3HRL`]:** We added detailed comparisons in **Appendix E** demonstrating why aggressive entropy minimization causes degenerate repetition, and how our methods balance perplexity reduction with diversity preservation.

- **Budget-Efficiency Discussion [Reviewer `Xwb4`]:** We clarified that our contribution provides an internal quality metric enabling effective selection from multiple candidates, addressing the core challenge in MDM decoding.

### 4. Additional Results

- Added **P2** baseline results to **Table 2** [Reviewer `TKiw`]
- Expanded discussion on distributional alignment vs. discrete task accuracy [Reviewer `TKiw`]
- Corrected typos throughout the manuscript [Reviewer `3HRL` `nHrU`]

---

We believe these revisions have substantially strengthened our work. We are grateful for the reviewers' guidance.

---

### Meta-Review · Area_Chair_uXHS · 2026-01-07

**Summary:**

Reviewers generally agree that the problem is important and that the paper is technically sound, but raise concerns regarding (i) the strength and novelty of the proposed entropy formulation relative to prior work, (ii) the validity and realism of the theoretical assumptions, (iii) whether empirical gains can be disentangled from increased computational budgets, and (iv) the practical significance of the observed improvements given the added inference-time complexity.

The rebuttal and revision substantially improve clarity, add missing experiments (runtime, diversity, stronger evaluators), and refine theoretical framing. However, several core concerns—especially about contribution novelty and budget-fair evaluation—are only partially resolved.

**Reviewer Concerns:**

Concerns largely addressed by the rebuttal include stronger theoretical clarity and assumptions, evaluation with stronger models, and a clarified experimental design. The following concerns are only partially addressed or remain outstanding:

- **Strength of novelty (gains) relative to prior methods:**
It remains unclear whether the proposed method constitutes a fundamentally new contribution rather than a repackaging or aggregation of existing work. Moreover, improvements on tasks other than Countdown appear marginal and the improvements may be modest given the added complexity. The authors argue that the novelty of the entropy notion lies in path-level uncertainty rather than token-level uncertainty, and that the gains should be interpreted in the context of the experimental design and the fundamental challenge of controlling MDMs. However, the rebuttal remains largely explanatory rather than demonstrative. As a result, skepticism about the incremental nature of the contribution persists.

- **Budget-fairness and disentangling algorithmic gains from sampling budget:**
 While the authors note that Pass@1 accuracy involves a selection problem and that Denoising Entropy provides a principled selection criterion absent in baselines, the response does not fully substitute for budget-matched comparisons under equal NFE or wall-clock constraints. Comparisons to majority voting help, but the lack of fully budget-fair baselines remains a valid weakness.

**Reviewer Scores:**

**Reviewer 3HRL (Score: 4, marginally below acceptance):**

Most concrete concerns were substantively addressed. This reviewer might be persuaded toward a slightly more positive stance, but the assessment would likely remain in the borderline range.

**Reviewer nHrU (Score: 4, marginally below acceptance):**

The clarifications address several technical points. However, the original strong assumption has only been refined, and concerns about uneven empirical gains are only partially resolved, suggesting that the score would likely remain around 4.

**Reviewer TKiW (Score: 6, marginally above acceptance):**

This reviewer was already positive and has stated that he/she will keep the score.

**Reviewer Xwb4 (Score: 2, reject):**

Core concerns about budget fairness and limited novelty remain unmitigated. While the authors provide arguments, a slight positive adjustment is possible, but the assessment would likely remain in the borderline range.

---

### Decision · Program_Chairs · 2026-01-26

Reject